# STRAP regulates alternative splicing fidelity during lineage commitment of mouse embryonic stem cells

Lin Jin[1,2], Yunjia Chen[3], David K. Crossman [3], Arunima Datta[1,2], Trung Vu[1,2], James A. Mobley[4], Malay Kumar Basu[5], Mariangela Scarduzio[6], Hengbin Wang[7], Chenbei Chang[8] & Pran K. Datta [1,2 ✉]

Alternative splicing (AS) is involved in cell fate decisions and embryonic development. However, regulation of these processes is poorly understood. Here, we have identified the serine threonine kinase receptor-associated protein (STRAP) as a putative spliceosome-associated factor. Upon *Strap* deletion, there are numerous AS events observed in mouse embryoid bodies (EBs) undergoing a neuroectoderm-like state. Global mapping of STRAP-RNA binding in mouse embryos by enhanced-CLIP sequencing (eCLIP-seq) reveals that STRAP preferably targets transcripts for nervous system development and regulates AS through preferred binding positions, as demonstrated for two neuronal-specific genes, *Nnat* and *Mark3*. We have found that STRAP involves in the assembly of 17S U2 snRNP proteins. Moreover, in *Xenopus*, loss of *Strap* leads to impeded lineage differentiation in embryos, delayed neural tube closure, and altered exon skipping. Collectively, our findings reveal a previously unknown function of STRAP in mediating the splicing networks of lineage commitment, alteration of which may be involved in early embryonic lethality in mice.

[1] Division of Hematology and Oncology, Department of Medicine, UAB Comprehensive Cancer Center, University of Alabama at Birmingham, Birmingham, AL 35294, USA. [2] Birmingham Veterans Affairs Medical Center, Birmingham, AL 35233, USA. [3] Department of Genetics, University of Alabama at Birmingham, Birmingham, AL 35294, USA. [4] Department of Anesthesiology and Perioperative Medicine, University of Alabama at Birmingham, Birmingham, AL 35294, USA. [5] Department of Pathology, University of Alabama at Birmingham, Birmingham, AL 35294, USA. [6] Department of Neurology, Center for Neurodegeneration and Experimental Therapeutic, University of Alabama at Birmingham, Birmingham, AL 35294, USA. [7] Department of Biochemistry and Molecular Genetics, University of Alabama at Birmingham, Birmingham, AL 35294, USA. [8] Department of Cell, Developmental, and Integrative Biology, University of Alabama at Birmingham, Birmingham, AL 35294, USA. ✉email: prandatta@uabmc.edu

Early mammalian embryogenesis is characterized by a series of cell fate decisions that initiate pluripotent cells to transit to cellular context-dependent lineage segregation, followed by progressive assembly of early organs. At gastrulation of the mouse embryo (E6.5–7.5), migrating epiblast cells give rise to the body pattern, and stem/progenitor cells are committed to three embryonic germ layers. The embryo undergoes organogenesis at early-somite stages (E8.0–8.5) and further establishes a more distinct embryonic configuration with a head, heart, limbs, and spinal cord as early as E8.75/9.0[1].

Using single-cell RNA sequencing (sc-RNA seq), researchers have profiled the transcriptome of mouse embryos at various stages of early development from one-cell stage to mid-gastrulation[2–4]. Recently, genome-wide epigenetic studies provide insight into patterns of epigenetic modulation and divergence in mouse preimplantation embryos[5,6]. However, in the early mouse embryo, the regulatory mechanisms at the post-transcriptional level for orchestrating germ layer determination and morphogenesis remain largely elusive.

For RNA transcripts, AS, a post-transcriptional event, results in substantial proteomic expansion and is conserved in a function-specific manner across vertebrate species[7]. The pre-mRNA splicing process requires the dynamic assembly of RNA-binding proteins (RBPs) into spliceosome machinery, a highly organized intra-nuclear structure consisting of RBPs and small nuclear RNA complexes (snRNAs), as well as other splicing factors, together called small nuclear ribonucleoproteins (snRNPs)[8,9]. The U1, U2, U4/U6, and U5 snRNPs are the main dynamic components of the major spliceosome, which is responsible for removing most pre-mRNA introns[10,11].

We determine that the WD40 domain-containing protein STRAP (also known as UNRIP[12]) promotes tumorigenicity[13] and maintains cancer stem-like cells[14]. STRAP is involved in intracellular distribution of the survival motor neuron (SMN) complex[15,16] and in regulation of cap-independent translation of viral mRNAs[12]. Besides these, to our knowledge, nothing is known about the function of STRAP in AS during development. Here, we show that STRAP interacts with components of U2 snRNP and that its deficiency affects the splicing fidelity in neuroectoderm-like cells. eCLIP-seq reveals that STRAP associates with a broad set of transcripts involved in nervous system development. By use of Xenopus embryos, we have found that AS sites recognized by STRAP are at highly conserved nucleotide sequences throughout evolution. Thus, our study deciphers the role of STRAP in modulating splicing programs associated with lineage-specific commitment.

## Results

**Substantial AS events occur during mouse early organogenesis.** To delineate molecular features of mouse embryos from post-gastrulation (E8.0) to early organogenesis (E9.0), we profiled global transcripts expression using whole mouse embryos at these two stages. At the E9.0 stage, there were substantial changes in transcripts, including signature genes involved in the formation of specialized organ systems (Supplementary Fig. 1a). In contrast, key regulators of early development were enriched at the E8.0 stage (Supplementary Fig. 1a). We thus identified groups of differentially expressed targets at the onset of embryonic organ formation.

Moreover, we identified 896 genes differentially expressed at the isoform level (Supplementary Data 1), as exemplified by Etl4, Nin, and Lmna (Supplementary Fig. 1b), suggesting that variants of genes contribute to transcriptional diversity in a developmentally regulated manner. To understand AS patterns in the transitional stage (from E8.0 to E9.0), we performed percent spliced-in (PSI) analyses of AS using the rMATS tool[17]. We obtained 1264 AS events for 1035 protein-coding genes, including skipped exon (SE), mutually exclusive exon (MXE), alternative 5′ splice site (A5′SS), alternative 3′ splice site (A3′SS), and retained intron (RI) categories (Fig. 1a, b and Supplementary Data 2). The regulated AS events mainly distributed in SE and MXE (38% and 41.4%, respectively) (Fig. 1b). Further, their PSI values had a uniform distribution between enhanced and repressed splice junctions (Fig. 1c). Gene Ontology (GO) function of AS genes revealed diverse enriched ontologies, including actin binding, cell division, brain and heart development, as well as other housekeeping cellular processes (Fig. 1d). We also performed RT-PCR to confirm several AS changes (Fig. 1e), finding a positive correlation with PSI values derived from RNA-seq (Supplementary Fig. 1c). Collectively, we uncover previously uncharacterized AS signatures during mouse early organogenesis and established a reference dataset for mammals.

**STRAP has a regulatory role in mouse early organogenesis.** We recently employed zinc finger nuclease (ZFN) technology to generate Strap deficient mice[18]. Although all Strap[+/−] male and female mice did not display any phenotypic abnormalities, their intercrosses yielded no homozygous offspring. We then collected embryos at selected days of gestational development to determine the stage at which Strap[−/−] embryos die. At E7.5 and E8.5, there were no obvious differences in embryonic morphology between wild type (WT) and mutant embryos (Fig. 2a). However, compared to WT littermates, E9.5 Strap[−/−] embryos were smaller with delayed development, namely dilated heart cavities and no body turning with truncated frontonasal regions (Fig. 2a), consistent with a previous report of gene trap mutagenesis[19]. However, that study reported only morphological changes; no mechanism was provided for embryonic lethality at cellular or molecular levels. Although E10.5 Strap[−/−] embryos were incompletely resorbed, E11.5 or older Strap[−/−] embryos were not obtained (Fig. 2b), indicating that one functional Strap allele is essential for mouse embryonic development and that, after E9.5, a double mutant leads to embryonic lethality. To study the spatial expression pattern of Strap in mouse embryos, a whole-mount in situ hybridization (WISH) assay was performed using E9.25–9.5 mouse embryos. Although STRAP was expressed throughout the body, its expression was stronger in the developing brain, eyes, limb buds, and neural tube (Supplementary Fig. 2a).

Next, we delineated the molecular cause of the Strap[−/−] lethal phenotype. At E7.5, the expressions of primary germ layer markers, including Sox1, Brachyury T, and Cer1, were not altered upon Strap deletion (Supplementary Fig. 2b). Strap[−/−]mutants, however, failed to express several early forebrain and midbrain developmental markers (i.e, Fgf8, Gsc, Otx2, and Shh[20,21]) at E8.5, (Fig. 2c). Other early brain markers (i.e, En1 and Hoxb1 for early hindbrain[22,23] and Six3 for the rostral forebrain fate[24]), and early cardiac and endoderm lineage markers (Gata4 and EpCAM) were not affected by Strap ablation (Supplementary Fig. 2c, d). In E9.5 Strap[−/−] embryos, expression of the later germ layer[25–27] markers was lower as compared to those in either Strap[+/+] or Strap[+/−] counterparts (Fig. 2d). Together, these results indicate that Strap deficiency initially has a negative impact on mouse embryo forebrain and midbrain development at E8.5 and subsequently causes a uniform and constant developmental delay from E9.5. Immunohistochemical analysis also revealed that NESTIN was largely absent from primitive neuroepithelia tubules in Strap[−/−] teratoma compared to that in WT, characterizing an immature neural tissue lacking neural progenitors (Fig. 2e). However, the expression of other two germ layer markers was comparable between groups (Fig. 2e).

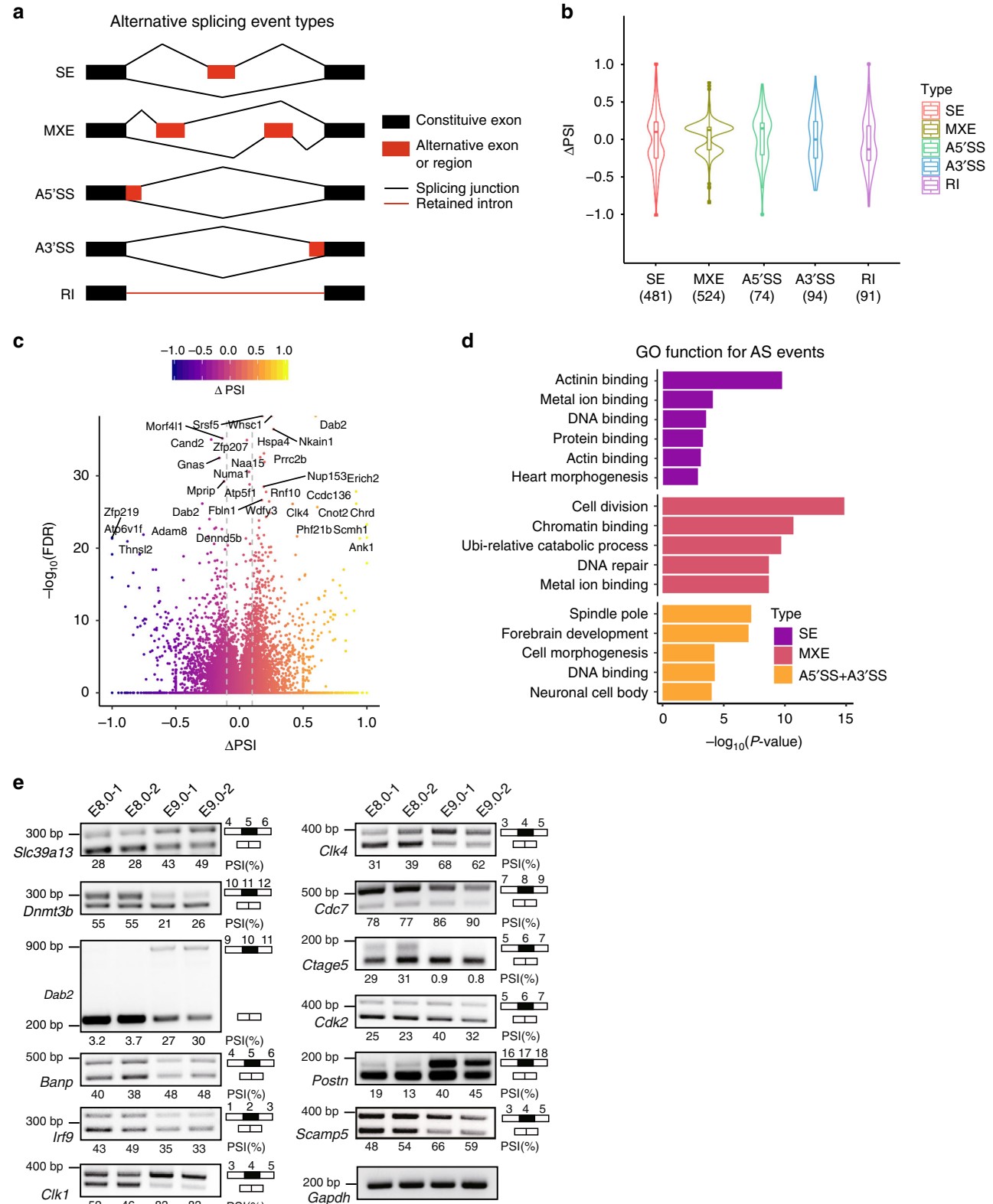

**STRAP interacts with components of spliceosomal machinery.** STRAP is linked to the cytosolic and nuclear SMN complex[15,28], but its other functions in the nucleus have not been elucidated. To determine this, we performed co-immunoprecipitation (co-IP) experiments with antibodies against STRAP and IgG using nuclear extracts (NEs) from mouse ESCs (Supplementary Fig. 3a). Co-IP samples were then visualized on SDS-PAGE and identified by LC-MS/MS (Supplementary Fig. 3b and Supplementary Data 3). Multiple sub-complexes of the spliceosome were identified, including components of the major snRNPs (U1, U2, U4/U6, and U5) and some splicing factors (Fig. 3a). The ranked scores assigned by the COMPLEAT tool[29] showed that 29.6% of hits (45 of 152) participated in various predicted networks between splicing protein interactions (Fig. 3b and Supplementary Fig. 3c).

**Fig. 1 Mouse embryo early organogenesis is associated with transcript isoform diversity. a** Schematic display of five AS types. SE, skipped exon; MXE, mutually exclusive spliced exon; A5′SS/A3′SS, alternative 5′/3′ splicing site; RI, retained intron. **b** Violin plots representing distributions of statistically significant ΔPSI (percent spliced-in) values (ΔPSI = PSI(E9.0)–PSI(E8.0); |ΔPSI|> 0.1, FDR < 0.05) for different classes of AS events. Kernel density is shown as a symmetric curve. The lower and upper bounds of the embedded box represent the 25th and 75th percentile of the distribution, respectively. The horizontal line in the box represents the median. The lower and upper whiskers show minima and maxima, respectively. The numbers of events are shown below each plot. **c** Volcano plot showing the difference of AS events between E8.0 and E9.0 mouse embryos. ΔPSI is plotted against the $-\log_{10}$(FDR) value. The color points outside the two dashed gray lines in the plot represent the differentially expressed AS with statistical significance (|ΔPSI|> 0.1, FDR < 0.05). Genes with $-\log_{10}$(FDR) >= 20 are indicated. **d** GO analysis of alternatively spliced genes between E9.0 and E8.0 mouse embryos, showing the top five to six ranked terms. **e** Validation by RT-PCR for genes with SE events identified by rMATS. PSI values are shown below the gel pictures. The information of target exons is shown on the right panel. Empty box, constitutive exon; black box, skipped exon. The experiment was repeated three independent times with similar results. *Gapdh* was used as a loading control.

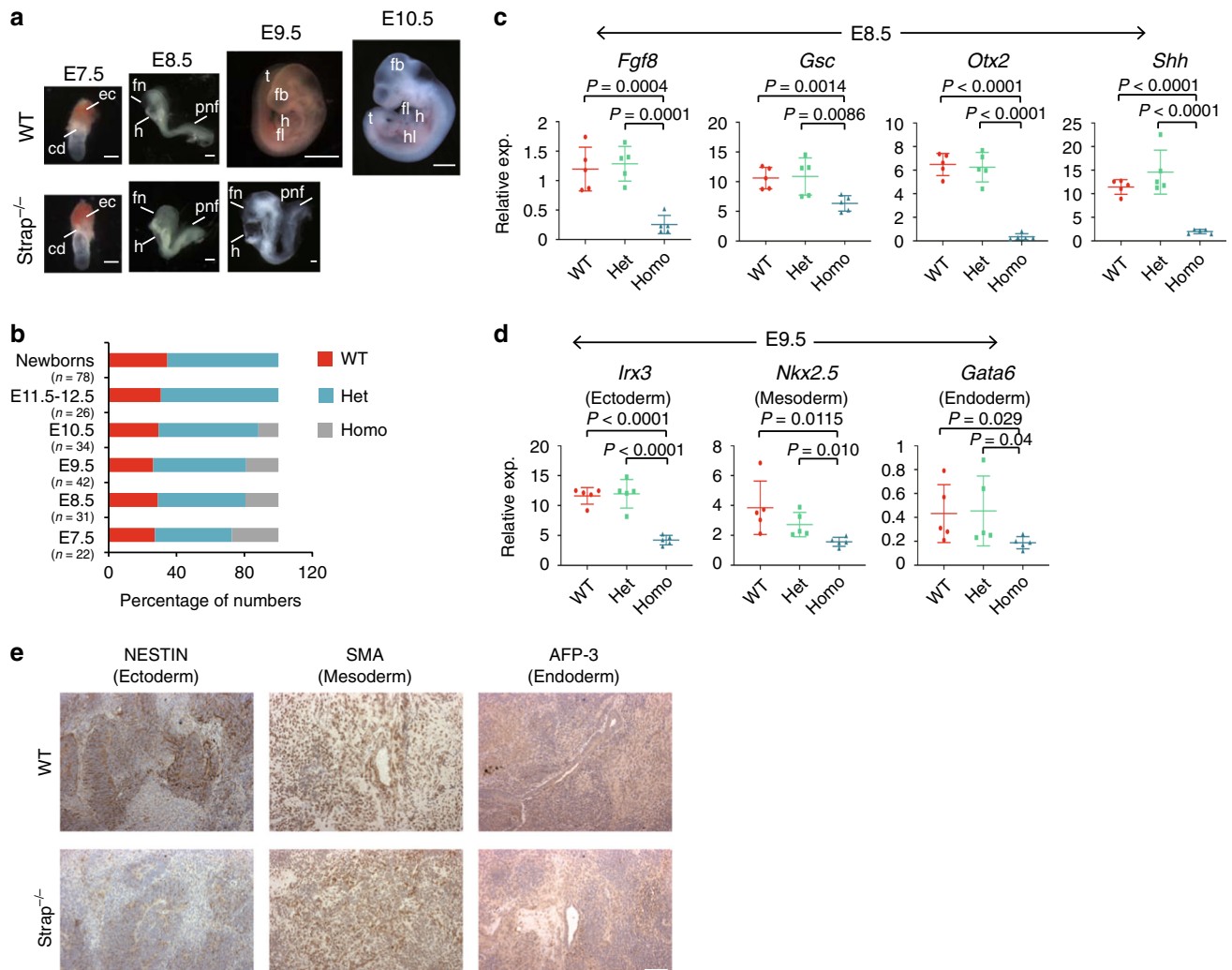

**Fig. 2 Genetic deletion of *Strap* causes mouse embryonic lethality. a** Representative litters of E7.5–10.5 embryos from intercrosses of B6 *Strap*$^{+/-}$ mice. Compared with a wild-type E9.5, a *Strap*$^{-/-}$ embryo shows size and morphological defects. For E7.5–E8.5 embryos, the experiment was repeated five independent times with similar results. For E9.5–E10.5 embryos, fourteen independent experiments were repeated and had similar results. For E7.5–8.5 and E9.5 *Strap*$^{-/-}$ embryos, scale bar equals 300 μm. For E9.5 WT and E10.5 embryos, scale bar equals 1 mm. ec, ectoplacental cone; cd, chorionic dome; fn, frontonasal region; h, heart; pnf, posterior neural folds; fb, front brain; fl, front limb; hl, hind limb; t, tail. **b** Histograms showing the survival percentages of embryos or pups from three genotypes at the indicated time. **c, d** qRT-PCR was used to quantify the relative mRNA levels in E8.5 (**c**) and E9.5 embryos (**d**). *P*-values based on unpaired two-tailed Student's *t*-test. Error bars indicate the mean ± SD from *n* = 5 biological replicates. WT, wild type; Het, heterozygous; Homo, homozygous. Data represent one of three independent experiments. **e** Paraffin embedded tissue sections from mouse ESC teratomas were stained with three germ-layer markers as indicated. The experiment was repeated two independent times with similar results (*n* = 3 biological replicates per group). Scale bar, 50 μm. Magnification, ×40.

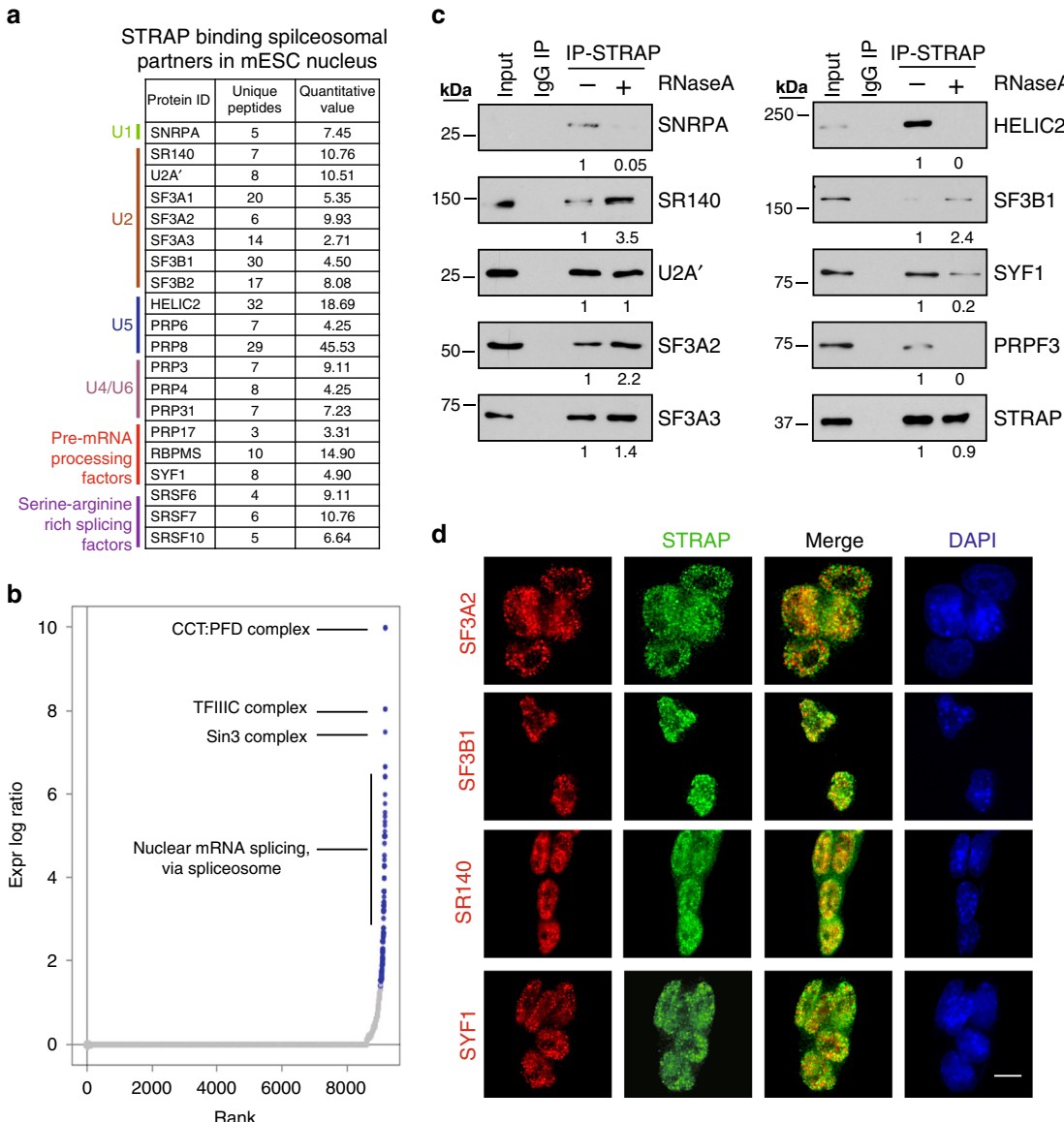

**Fig. 3 Intracellular STRAP binds with subunits of the spliceosome complex. a** List of STRAP-binding proteins for spliceosome complex subunits. Anti-STRAP antibody was used to immunoprecipitate STRAP and co-precipitate its interacting proteins from mouse ESC nuclear extracts (NE). The binding partners were identified by LC-MS/MS. **b** Ranked curve for STRAP binding-partners (see method). Enriched complexes (colored dots) are those that meet the P-value cutoff (P < 5.012e−2). Top enriched complexes are indicated. **c** Interactions between indicated proteins and STRAP were confirmed by co-immunoprecipitation and Western blotting analyses. Samples were prepared as in (**a**) and treated with or without ribouclease A (RNase A). 1% of lysates were loaded as input control. The levels of co-precipitated proteins after RNase treatment (relative to untreated control) are shown below the gel pictures. The experiment was independently repeated twice with similar results. **d** The localization of STRAP (green) and indicated proteins (red) in mouse ESC cells were analyzed by immunofluorescence. Cell nuclei were labeled by DAPI (blue) staining. Scale bar, 10 μm. Images were captured by a Keyence microscope. The experiment was repeated three times independently with similar results.

The interactions between STRAP and some of the above-mentioned partners were confirmed by probing same immuno-precipitated NEs using indicated antibodies (Fig. 2c). Treatment with RNase led to unchanged or enhanced the association of STRAP with U2 proteins (such as U2A', SF3A/3B subunits, and SR140), suggesting protein–protein interactions. There was no interaction, however, between STRAP and spliceosome disassemble protein DDX15 (Supplementary Fig. 3d). Of note, there was pull-down of TFIIIC110 and STRAP in the absence of DNase (Supplementary Fig. 3e), indicating that STRAP acts as a bridge to link the complex of protein(s) and co-transcriptional RNA:DNA hybrid structures. We will explore this possibility in future work. Immunofluorescence staining assays confirmed that STRAP co-localized with various spliceosomal components in the nucleus (Fig. 3d). These results suggest a function of STRAP in the spliceosomal complex.

**STRAP-mediated AS events are involved in mouse early neuroectoderm lineage commitment.** Since loss of STRAP first affects expression patterns of early brain region-specific genes in vivo, we established N2B27-induced EB models to direct ESCs into neuroectodermal fate[30,31] and, later, to neuronal cell identity[32]. The numbers, size, and morphology of EBs derived from WT and *Strap* KO ESCs appeared similar over time

(Supplementary Fig. 4a). In addition, the expression of STRAP was steady during differentiation of WT EBs (Supplementary Fig. 4b). To define the molecular features of developing EBs, quantitative PCR (qPCR) assays were performed over the course of time. At day 6-9, expression of neuroectoderm-related genes was comparable between the two groups, (Supplementary Fig. 4c). Brain developmental markers appeared as early as day 9, but, at day 11, most of them were low in *Strap* KO EBs as compared with WT counterparts (Supplementary Fig. 4d), consistent with in vivo studies (Fig. 2c). Together, the results suggest that STRAP functions during early neural differentiation. We thus conducted RNA-sequencing using 9-day-old EBs, in which WT and KO cells commit to the similar lineage. Global comparisons using principle components analysis (PCA) and hierarchical clustering analysis revealed that EB replicates in the same group had high reproducibility, and that their identity was close to previously reported neuroectoderm organoids[33] than to other developmental derivatives[34,35] (Supplementary Fig. 4e, f). Therefore, our N2B27-induced EB model molecularly resembles the mouse embryonic germ layer specification and provides us an ideal model to dissect the role of STRAP in mouse early embryogenesis.

We next characterized transcripts and splicing variants with genome-wide profiling. Only 154 genes were altered in transcriptome profiling of *Strap*-KO ESCs relative to that of ESC WT cells (Supplementary Data 4), suggesting that STRAP is dispensable for mouse ESCs viability and identity[18]. Loss of STRAP in 9-day-old EBs resulted in 146 genes upregulated and 261 genes downregulated (Supplementary Data 4). As determined by use of the rMATS tool, 454 classical AS events (in 397 genes) were detected in *Strap*-KO EBs at day 9 (Fig. 4a and Supplementary Data 5). Of all AS, SE events were most frequent (49.1%, 223 of 454) (Fig. 4a, b). Retained introns were also overrepresented (18.1%, 82 of 454) as compared to other types (Fig. 4a). Thus, STRAP appears to function in promoting both exon inclusion and skipping (Fig. 4a), implicating its unbiased regulation for SE. GO function analysis revealed that STRAP-regulated AS events are involved in protein catabolism, molecular binding, and cell motility (Fig. 4c). Specifically, the events included neural development, ubiquitin-related genes, and transcription factors (Supplementary Fig. 4g–j). To assess the accuracy of our analysis of AS, we measured 28 SEs using 9-day-old EBs (Supplementary Fig. 4k), demonstrating a high correlation with splicing efficiency (Supplementary Fig. 4l).

By comparing splice sites scores in responsive and unresponsive SEs, we found that both STRAP-enhanced and STRAP-repressed exons have weaker 3′ splice site (ss) than those found in unresponsive SE (unpaired Wilcoxon test, Fig. 4d, e). In contrast, we observed unbiased strength in either the 5′ss of SE or their flanking sites (Fig. 4d, e). To examine how STRAP correlates with other splicing factors in terms of the splicing level, we used a public dataset[36] and compared the properties with AS genes between STRAP and several core reported AS factors. STRAP was well correlated with SF3 family members and PRPF3 (Fig. 4f), suggesting the potential cooperation of STRAP with these proteins. Only a few AS outcomes were affected by deletion of STRAP, either in mouse ESCs (Supplementary Data 6, $n = 130$, EB vs. ESC, $p < 0.0001$, Fisher's extract test) or in mouse intestinal-specific tissues (Supplementary Data 6, $n = 210$, EB vs. cKO, $p < 0.0001$, Fisher's extract test), suggesting that STRAP mediates AS in a cell-type-specific manner.

To explore overall mRNA isoforms regulated by STRAP during neuroectoderm differentiation, we reanalyzed RNA-seq data using the MAJIQ (modeling alternative junction inclusion quantification) tool[37] to quantify both classic binary AS events and complicated local splicing variants (LSVs). We obtained 1462

altered LSVs (of 1163 genes) upon deletion of STRAP in 9-day-old EBs ($|\mathrm{dPSI}| >= 0.1$, $p < 0.05$, Supplementary Data 7), as exemplified by *Dnmt3b* and *Zkscan1* (Fig. 4g). Moreover, these results had a significant concordance with data from rMATS (Fig. 4h). Thus, these analyses revealed a previously unknown role of STRAP involving the transcriptional diversity at exon levels during a certain germ layer differentiation.

**STRAP displays genome-wide mRNA binding patterns**. We next asked whether STRAP binds to its target RNAs in vivo. To this end, we utilized eCLIP-seq, which yields high complex libraries at single nucleotide resolution with enhanced technical and biological reproducibility[38]. Radiolabeled STRAP-bound RNA complexes from mouse embryos and EBs were visualized (Fig. 5a). Stringent RNase I treatment markedly eliminated STRAP-RNA bands (Fig. 5a), suggesting that the robustness and specificity of the STRAP-RNA complexes identified by eCLIP. As shown in Supplementary Fig. 5a, reads density was highly correlated across replicate samples, suggesting that these STRAP-binding clusters were due to intrinsic biological functions instead of technical variations. We obtained, in total, 9686 STRAP high-confidence peaks located at 2047 individual transcripts on a genome-wide scale (Supplementary Data 8). Most STRAP-peaks were defined by certain exons and introns, as well as their intersections (Fig. 5b and Supplementary Fig. 5b). We next leveraged STRAP reads along all exon/intron (5′ss) and intron/exon (3′ss) borders. In comparison to the input, STRAP signals in embryonic samples were enriched within ~50 nucleotides upstream of the 5′ss and ~25 nucleotides downstream of the 3′ss (Fig. 5c). Similar binding patterns were also evident in EBs (Fig. 5d). Thus, the binding of STRAP in close proximity to both 5′ and 3′ of splicing sites suggests its role in demarcating exonic regions. In addition, these featured occupancies were largely dependent on their transcript abundance (Supplementary Fig. 5c), in agreement with a previous report for CLIP-seq data[39]. Notably, among these exonic hits, 964 peaks appeared on annotated alternative exons and 2112 peaks were associated with constitutive exons (within the 300 nt region either upstream or downstream of exons, Supplementary Data 9), thus providing evidence for its involvement of STRAP in the AS process.

To identify the binding site consensus sequences for STRAP, we mapped STRAP binding peaks and identified the top-ranking enriched motifs (Fig. 5e). The fractions of targeted sequences were comparable (Fig. 5e, % of Targets), implying that STRAP cooperates with other RNA-processing regulators to recognize different types of sequences. To verify the direct binding of STRAP with RNAs, we conducted RNA electrophoretic mobility shift assays (REMSA) using a probe targeting the 3′ UTR of the *Nnat* transcript, which encompasses two putative STRAP consensus binding sequences (Fig. 5e). STRAP binds to the WT sequence but not to the mutant (MUT) probe (Fig. 5f). The addition of excess cold probe competed effectively for binding during the assay (Fig. 5f). These results demonstrate the RNA binding specificity of STRAP. To illustrate the biological functions of the loci related to STRAP peaks, we performed GO analysis, revealing genes for nervous system development, cytoskeleton, centrosome, and spliceosomal complex (Fig. 5g). Examples of STRAP binding peaks for neural transcripts are shown in Supplementary Fig. 5d-f. These data suggest that transcriptome-wide binding sites are differentially bound by STRAP during early development of mouse embryonic brain.

**STRAP dynamically regulates neuronal exon splicing during EB differentiation**. We next determined if STRAP binding is linked to its regulated genes at AS levels. Using the annotated

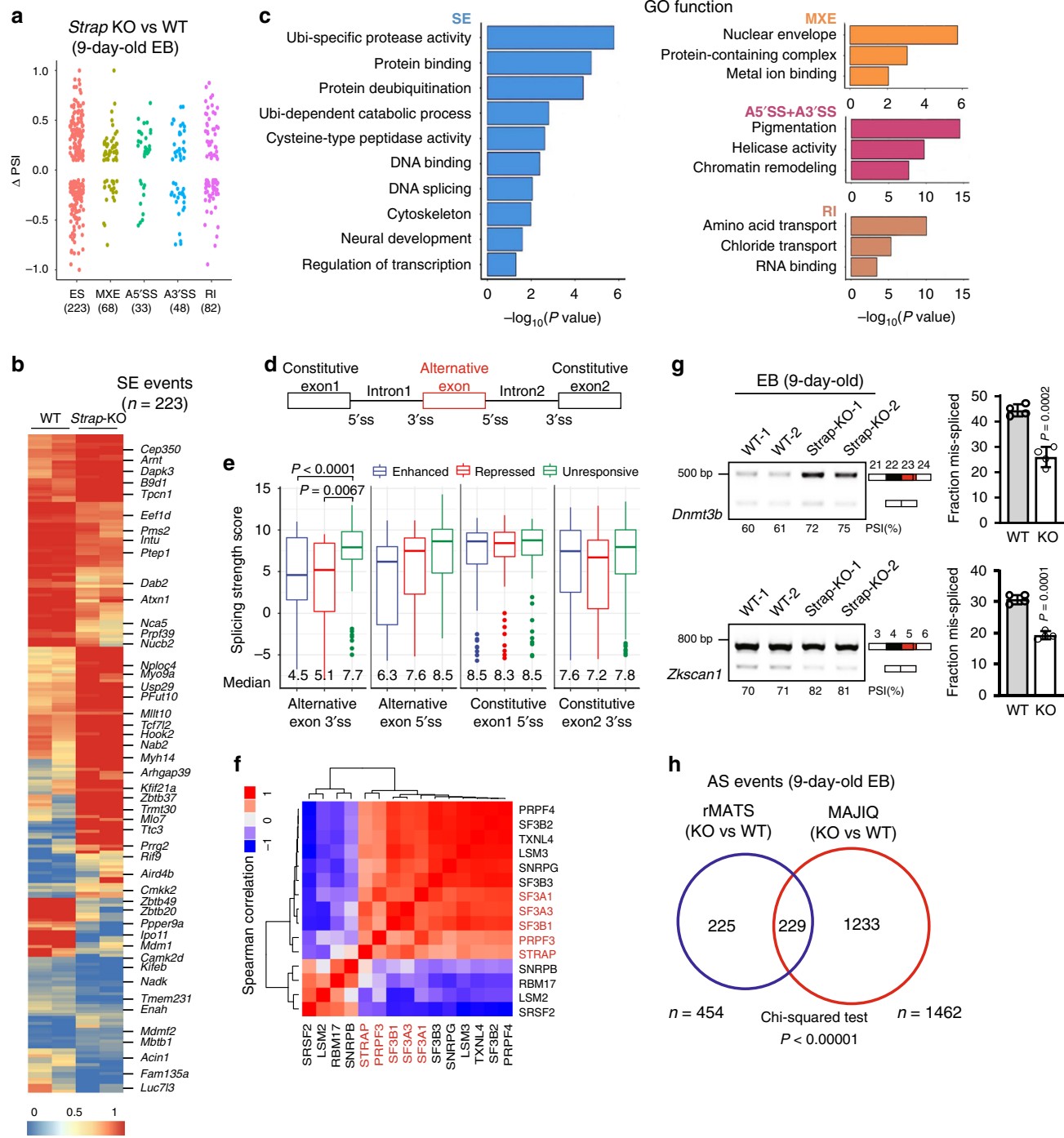

genes list derived from merged eCLIP peaks as a reference dataset, we found that 15.6% (319 of 2047) of STRAP binding events was associated with its regulated AS genes (both classic types and non-classic types) and that 22.4% (319 of 1423) of unique genes with AS events in 9-day-old EBs was linked to STRAP binding (Fig. 6a), supporting our hypothesis that STRAP developmentally regulates certain AS events through direct binding. These results prompted us to assess the distribution of STRAP-RNA interactions in STRAP-dependent AS events. We thus mapped STRAP binding peaks to alternative exons and their flanking constitutive exons, as well as surrounding intronic regions. Figure 6b shows a general increase in STRAP occupancy surrounding the 3′ss of STRAP-enhanced exons and neighboring

intronic regions, in addition to 5′ss of upstream flanking exons. Similar trends were also evident in STRAP-repressed exons (Fig. 6c), indicating that STRAP collaborates with other splicing regulator(s) to cause the final splicing outcomes. Furthermore, we performed UV crosslinking and immunoprecipitation (RIP) with anti-STRAP antibody to examine three eCLIP-hit genes, *Tcf7l2*, *Ctnnd1*, and *Fat1*. qPCR profiles further confirmed that STRAP was associated with these targeted transcipts (Supplementary Fig. 6a–c). Upon deletion of STRAP, there was no change in their transcripts (Supplementary Fig. 6d–f), ruling out the possibility that STRAP interacts to target RNAs due to their transcriptional abundance. Together, these results indicate that STRAP selectively binds AS regions for their processing.

**Fig. 4 Extensive AS events occur in response to *Strap* loss in lineage-committed EB cells. a** Dot plot showing distribution of ΔPSI values (ΔPSI = PSI (*Strap*$^{-/-}$)−PSI(WT)) for each splicing category. The value of ΔPSI between WT and *Strap*$^{-/-}$ EBs was based on two independent biological samples from each group. **b** Splicing heatmap showing the values of PSI for exon-skipping events between WT (*n* = 2, biological replicates) and *Strap*$^{-/-}$ (*n* = 2, biological replicates) EBs. **c** GO enrichment analysis showing top ranked biological functions for alternatively spliced genes between WT and *Strap*$^{-/-}$ EBs (9-day-old). **d** A schematic of the alternative and constitutive exons. 5′ 3′ splicing sites (5′,3′ ss) are indicated accordingly. **e** Box plots of splice site scores calculated for STRAP-regulated skipped exons. Boxplots show median (the horizontal line in the box), 25 and 75% percentiles (lower and upper bounds of box, respectively), minimum and maximum (lower and upper whiskers, respectively). *P*-values were determined by two-sided unpaired Wilcoxon tests. For ALT3′ss, unresponsive *n* = 79, enhanced *n* = 61, repressed *n* = 53; for ALT5′ss, unresponsive *n* = 67, enhanced *n* = 53, repressed *n* = 33; for CON1 5′ss, unresponsive *n* = 67, enhanced *n* = 73, repressed *n* = 53; for CON2 3′ss, unresponsive *n* = 73, enhanced *n* = 82, repressed *n* = 63. **f** Hierarchically clustering matrix showing the correlation for STRAP and other known splicing factors. Spearman correlation coefficients were calculated based on the ΔPSI values of skipping exons affected by either STRAP or previously reported splicing factors[36]. Highly correlated factors are highlighted in red. **g** Validation by RT-PCR of genes with SE events identified by MAJIQ. Information for target exons is shown on the right panel. Empty box, constitutive exon; black and red boxes, skipped exon. Data are pooled from two independent experiments (*n* = 4 per group) and error bars represent the mean ± SD (right panel). Unpaired two-tailed Student's *t*-test was used. **h** Venn diagram showing the intersection of AS events between rMATS and MAJIQ datasets. The detailed AS category and respective numbers are shown at the bottom. *P*-values were determined by Chi-squared test.

The WT EB model underwent neuronal differentiation after long-term N2B27 culture with ascorbic acid, as evidenced by the high percentage of CD24+/CD56+ neuronal cells at day 14 (Supplementary Fig. 6g) and gradually increased neurogenesis markers over time (Supplementary Fig. 6h). Compared to a WT parallel, KO EBs had lower levels of the CD24+/CD56+ subpopulation and unaltered expression patterns for selected neuronal genes (except for *Pax6*) along the differentiation (Supplementary Fig. 6g, h), indicating that EBs failed to undergo neuronal lineage upon deletion of *Strap*. We further assessed the terminal differentiation of EBs by assessing their electrophysiological properties. In whole-cell patch clamp recordings, we observed a small percentage of KO cells (20%) capable of producing immature single action potentials (APs) upon depolarization and a prominent sag upon hyperpolarization at day 12 compared to 40% of WT cells (Fig. 6d, e). In both genotypes, the percentages of cells capable of producing APs and sags increased with time, reaching 85% in WT vs. 25% in KO at day 22 (Fig. 6d, e). None of the cells in this phase of maturation produced trains of APs in response to depolarizing steps. However, a few WT cells at day 22 showed single or multiple "rebound" APs and spontaneous AP firing at resting membrane potentials (RMP) upon termination of hyperpolarizing pulses (examples in Fig. 6d, middle and bottom right panel, respectively), indicating a faster and more pronounced course of neuronal differentiation in WT than in KO cells.

During maturation of neuronal cells, stage-specific-switch exons in genes have distinct functions[40]. We intersected eCLIP genes with rMATS-derived classic SE genes (FDR < 0.1) and thus found 98 hits (Fig. 6f). Here, we investigated exons from two murine genes, *Nnat* and *Mark3*. AS of neuronatin (*Nnat*) generates two spliced isoforms, depending on the usage of exon2[41]. The short mutant has a more potent role in neural patterning than the full-length one[42]. In *Nnat*, extensive STRAP-binding sites located on exon1 and exon3 (including 5′ and 3′ UTRs, respectively) were detected by use of embryonic samples (Fig. 6g). RIP-qPCR validated that STRAP peaks on 14-day-old WT EBs are strongly present on the indicated boundaries as compared to 11-day-old parallels (Fig. 6h). Little enrichment was evident in other regions (Supplementary Fig. 6i). These interactions were not caused by RNA abundance (Supplementary Fig. 6k). We also observed a lower exon 2 inclusion in WT EBs compared to its earlier parallels and KO groups (Fig. 6i, j), suggesting that STRAP temporally regulates exon2 exclusion in *Nnat*.

Another example is *Mark3*, which encodes a kinase involved in the phosphorylation of MAP2 and MAP4[43]. Two spliced variants of *Mark3* are differently expressed in neural progenitors (exon16

inclusion) and neuronal cells (exon16 skipping)[44]. In embryos, accumulated STRAP-binding peaks are located on *Mark3* exon15, as determined by eCLIP (Fig. 6k). The transcripts with target sites were confirmed by RIP in 14-day-old WT EBs (Fig. 6l and Supplementary Fig. 6j), which were also coupled with more exon16 skipping relative to other parallels (Fig. 6m, n). Again, RNA products had no effect on these binding signals, as there were no altered *Mark3* transcripts on the indicated days (Supplementary Fig. 6l).

To eliminate the possibility that SE events were caused by differentiation arrest rather than STRAP-dependent binding regulation, we developed a Tet-On inducible short hair RNA (shRNA) system against STRAP in mouse E14 ESC cells. Three-day treatment of Doxycycline (Dox) resulted in an efficient reduction of STRAP expression in 14-day-old EBs derived from E14 cells (Fig. 6o, left bottom). Compared with control groups, induced knockdown of STRAP produced more included exons for *Nnat* and *Mark3* transcripts (Fig. 6o), consistent with observations with the KO EBs. These data suggest that, during EB differentiation, STRAP is involved in the selection of neural exons in a local concentration-dependent manner.

**STRAP is involved in the assembly of 17S U2 snRNP proteins.** As the RNase treatment did not interfere with STRAP binding to U2 snRNP components, indicating that these interactions depended on protein–protein contacts, we focused on the downstream effects of *Strap* knockdown. First, we measured the abundance of U2 proteins in whole-cell lysates extracted from ESCs and EBs and found higher levels of SF3A subunits and SR140 in STRAP KO 14-day-old EBs relative to WT parallels (Fig. 7a, b), suggesting that STRAP is involved in balancing homeostasis of U2 proteins at a specific stage. Similar trends were observed for Tet-On inducible shSTRAP EBs at day 14 (Supplementary Fig. 7a), ruling out the indirect effects caused by STRAP-KO impaired differentiation.

We next addressed the question of whether STRAP deletion affects formation of the U2 complex. Among U2 interactors, the SF3B complex is initially recruited to 12S core particles (the U2 snRNA, the Sm core proteins, and the U2A′ and U2B′′ complex) to generate pre-cmature 15S complex, which subsequently incorporates SF3A subunits to form the mature 17S snRNP[45–48]. Lastly, the functional U2 snRNP is recruited to 3′ss of the intron, together with the U1 complex at the 5′ss to form pre-spliceosomes. To determine whether STRAP has effects on the assembly of the 17S U2 complex, we accomplished co-immunoprecipitations (co-IP) using an SF3B1 antibody on NEs from the indicated cells. In ESCs, interactions between U2 components were independent of STRAP (Fig.7c, right panel). However, during EBs differentiation

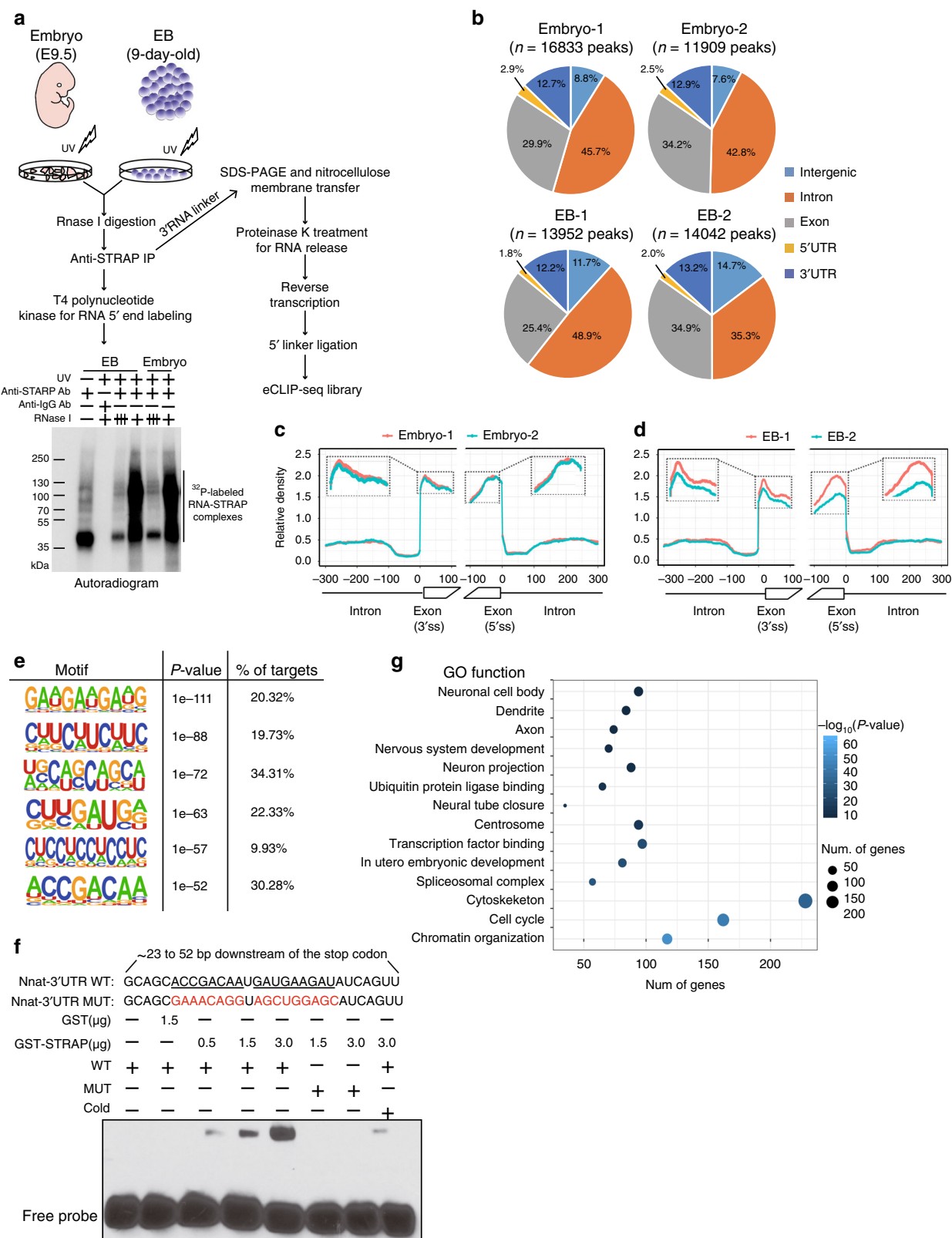

loss of STRAP led to a dissociation between SF3B1 and two SF3A subunits, whereas the interaction between SF3B1 and U2A' was independent of STRAP expression (Fig. 7c, right panel; 7d). Together, the results suggest that STRAP developmentally regulates assembly of the 17S complex. We also tested the interaction between SR140 and CHERP, a binding partners among 17S U2 components. Although there was no change in CHERP expression

(Fig. 7e, left panel), SR140 was less co-precipitated with CHERP in KO EBs relative to the WT group (Fig. 7e, right panel; 7f).

To map the binding domain(s) of STRAP responsible for its function, we generated deletion mutants of STRAP with Flag epitope tag and performed co-IP assays with HA-tagged SR140. The results revealed that the third WD-40 domain is necessary for the binding of STRAP with SR140 (Fig. 7g). To test whether

**Fig. 5 Discovery of in vivo STRAP-binding RNA targets by eCLIP-seq. a** Schematic illustration of the overall experimental design, showing that STRAP binds with $^{32}$P-labeled RNAs (Bottom); the eCLIP method was used to establish sequencing libraries (Right). The schematic diagram was created by the authors. **b** Pie charts displaying peak distributions of enriched read density within STRAP eCLIP. The fraction of STRAP peaks, defined from eCLIP-seq ($P \le 10^{-3}$ and $\ge$ 5-fold, relative to INPUT), locates along the different genic regions across the mouse transcriptome. Each biological replicate shows a highly comparable percentage for indicated categories between replicates. **c**, **d** Metagene plots showing STRAP binding frequency on pre-mRNA in two pooled embryos (E9.0; $n = 7$ per pool) (**c**) or two biologically replicated WT EBs (9-day-old) (**d**). The X-axis indicates a composite intro-exon-intron boundary, containing sequences for 300 nt in the upstream intron and the first and last 100 nt of the exon and 300 nt in the downstream intron. STRAP eCLIP crosslink site density around constitutive 5′ and 3′ splice sites normalized by respective input density is plotted on the Y-axis. The dash line boxes show amplified regions for peak enrichment. See "Methods" section for further details. **e** Left, logo visualization of the top HOMER motif outputs generated from the merged eCLIP dataset. Right, the fraction of target regions and respective P-value with each motif are displayed. **f** Binding of STRAP with *Nnat* motif-containing RNA oligonucleotides. Purified GST-STRAP or GST was incubated with biotin-labeled WT or MUT or 200-fold excess non-labeled RNA oligos as indicated. The complexes were separated on 6% polyacrylamide native gel. The experiment was repeated three times independently with similar results. **g** BubbleMap visualization of representative GO functions for STRAP target genes through RNA-protein interaction.

certain splicing variants could be regulated by the WD-40 domain (s) of STRAP, we performed gain-of-function assays in MEF cells using a mini-gene splicing tool, which contains *UPF3A* exon 4 in a SR140-dependent manner[49]. Compared to WT MEFs, the usage of exon 4 was significantly reduced in STRAP KO MEFs and could be partially rescued by overexpression of STRAP-Flag, STRAP (1–4)-Flag, and STRAP (3–4)-Flag mutants (Fig. 7h) and of SR140 (Supplementary Fig. 7b). Similar assays using other mini-gene tools in two gene contexts (*Nnat* and *Ppp2r2d*) also revealed that the third WD-40 domain of STRAP is essential for its splicing activity (Supplementary Fig. 7c, d).

**STRAP regulates exon skipping during vertebrate evolution.** The exon-intron structure is involved in the recognition of exon and intron length and affects exon inclusion levels[50–52]. We explored whether STRAP-dependent skipping exons have distinguishing sequence features. The lengths of enhanced exons are significantly shorter than those of controls (Supplementary Fig. 8a) and their upstream introns are longer length compared to controls (Supplementary Fig. 8b). Moreover, the repressed exons have significantly shorter upstream constitutive exons compared with controls (Supplementary Fig. 8c). No additional features correlated with other flanking regions (Supplementary Fig. 8d, e).

In our previous study, we found that STRAP affects neural patterning in early *Xenopus* embryogenesis[18]. Given that STRAP is a conserved WD-40 domain-containing protein across species, we hypothesized that it has a functional role in AS regulation in *Xenopus*. Thus, we injected a *Strap* antisense morpholino oligonucleotide targeting the translation start site (ATG-MO) into early *Xenopus* embryos to reduce the expression of STRAP (Supplementary Fig. 8f). In *Strap* morphants, there was impaired neural tube development, which was partially rescued by ectopic STRAP expression that could not be blocked by the MO (Fig. 8a, b). Further investigation of STRAP function in Noggin-neutralized ectodermal explants revealed that depletion of STRAP impaired neural ectoderm development, as evidenced by reduced expression of neural markers (Supplementary Fig. 8g). Moreover, STRAP evolutionarily controls AS in *Xenopus* neural tissues, as exemplified by the spindle microtubule-related gene *Cep57*[53] and the nuclear kinase gene *Clk1*[54] (Fig. 8c, d). RT-PCR assays showed that exon-specific skipping in these two genes was STRAP-dependent (Fig. 8d, e). However, total transcripts of these two genes had little change upon STRAP depletion (Supplementary Fig. 8h).

To explore the functional consequence of AS of STRAP targets, we focused on *Clk1*. Injection into dorsal animal region with MO against the *Clk1* splicing acceptor site at exon 4 (splice-MO) (Fig. 8f), which encodes a linker region immediately upstream of the protein kinase domain. This resulted in embryos displaying neural tube closure defects and severely impaired late

development of the head and body axis (Fig. 8g, middle panel). These defects were partially rescued by *Clk1* RNA (Supplementary Fig. 8i). Notably, this phenotype was different from that when *Clk1* ATG-MO was used. The ATG-MO induced severe gastrulation defects with the morphant embryos displaying exposed mesoderm at tailbud and tadpole stages (Fig. 8g, bottom panel). These results imply that, although complete blockage of both *Clk1* products impaired early *Xenopus* mesoderm development, selective alteration of the ratio of the two *Clk1* products led to neural development defects.

**Discussion**

In this study, we show that, for mice, the deletion of *Strap* leads to early embryonic lethality and, in EBs, to abnormal differentiation (Fig. 2 and Supplementary Fig. 4c, d), indicating an essential role of STRAP in mouse early embryo development and differentiation of ESCs. Since the AS program tightly controls the post-transcriptional genes necessary for cell differentiation and development, we reason that these defects are caused, in part, by AS events upon *Strap* KO. The following evidence supports this concept: (i) Substantial AS patterns, instead of limited transcriptional regulation, are affected upon *Strap* KO in lineage-committed murine ESCs; (ii) The involvement of STRAP for the assembly of the 17S U2 snRNP complex suggests its role in AS modification; (iii) STRAP contributes to the selection of important neuronal exons during EB differentiation, as exemplified by *Nnat* and *Mark3*; and (iv) *Strap*-MO in *Xenopus* embryos influences neural patterning and impaires the body axis[18].

Our work reveals that deletion of STRAP has effects mainly on exon skipping during the lineage commitment, which is distinct from the recently elucidated AS programs during the neural differentiation of human iPSCs, in which increased intron retention is a predominant feature of AS at early stage[55]. Among STRAP-mediated AS events, the binding typically takes place upstream of alternative exons (Fig. 6b, c), in agreement with the position-dependent splicing rule[39,56]. We also show the dynamics of STRAP enrichment on its RNA targets, thereby raising a possibility that it bridges spliceosome complexes mediating AS recognition and sequential catalysis during development.

Various RBPs preferentially bind low-complexity motifs comprised primarily of one or two base types[57,58], which is reflected in the abundance of GA-rich and CU-rich motifs for STRAP (Fig. 5e). A possible explanation is that these motifs facilitate cooperative binding of STRAP with other RBPs and enhance splice site recognition during exon ligation. In addition, distal intronic regions are reservoirs of highly conserved RNA cis elements necessary for splicing regulation[59]. Our RNA-binding map also reveals that some of STRAP peaks are located far from target exons (>200 nt). To date, the most commonly used models for protein-RNA binding and CLIP-seq experiments are stable cell

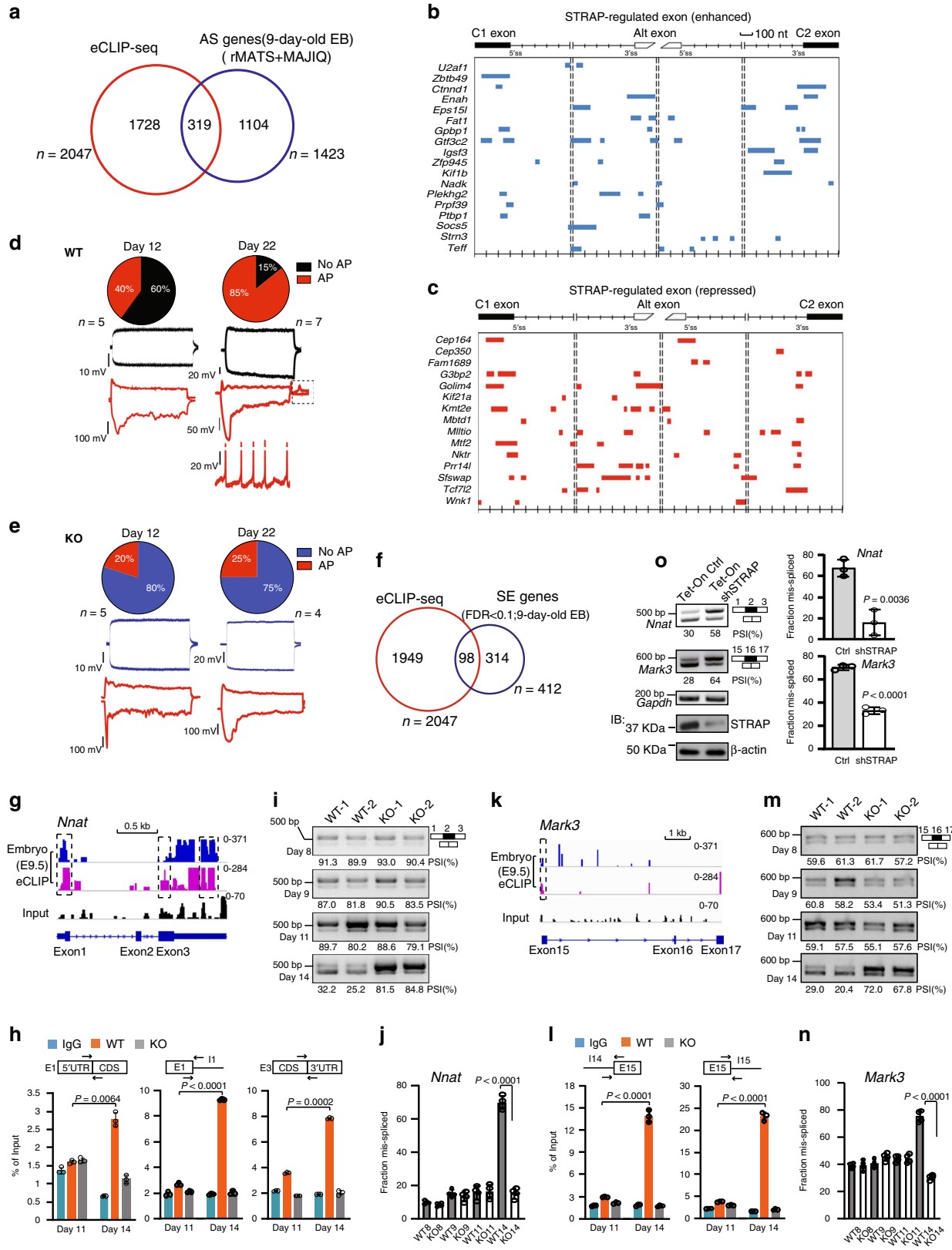

lines or regionally dissected tissues. These models, however, are unable to identify, temporally, RBP targets with stage-specific switch exons during cell differentiation. Here, we present an EB-based model, which facilitates understanding of the RNA processes during development.

In sum, these studies define a genome-wide AS network that STRAP regulates in a coordinated manner. We reveal the position-dependent and context-dependent effects of STRAP on splicing regulation during neuroectodermal lineage commitment, which adds another layer of complexity to the cross-

**Fig. 6 STRAP-RNA interaction map showing direct binding to skipped exons. a** Venn diagram showing the intersection of STRAP-regulated AS genes (defined by rMATS and MAJIQ) and genes within STRAP eCLIP peaks. **b**, **c** Composite maps for STRAP-binding peaks on AS events based on the RNA-seq data (|ΔPSI|> 0.1, FDR < 0.05). **d** At day 12, 40% of WT EBs showed an immature action potential (AP) in response to depolarizing current steps and a prominent sag in response to hyperpolarizing current step (red traces). At day 22, 85% of cells showed AP, potential sag, and "rebound" APs (dashed box in middle red traces). Some cells showed spontaneous AP firing at RMP (bottom red traces). **e** 20% of KO EBs showed immature AP at day 12 and 25% at day 22. **f** Venn diagram showing the intersection of STRAP-regulated SE genes defined by rMATS (FDR < 0.1) and genes within STRAP eCLIP peaks. **g** IGV viewer genome browser image of eCLIP signals on *Nnat* pre-mRNA. **h** RIP assay for STRAP binding on *Nnat* in EBs. Targeted regions (black outlined box) with respective paired primers are shown at the top. **i**, **j** RT-PCR quantified PSI values of *Nnat* in EBs. **k** IGV viewer genome browser image of eCLIP signals on *Mark3* pre-mRNA. **l** RIP assay for STRAP binding on *Mark3* pre-mRNA. **m**, **n** RT-PCR quantified PSI values of *Mark3* in EBs. **o** RT-PCR quantified PSI values of *Nnat* and *Mark3* in E14 cell line-derived EBs at day 14. Left bottom, Western blot analysis showing STRAP expression in inducible STRAP knockdown EBs with ß-Actin as a loading control. For (**i**, **m**), and (**o**), PSI values are shown below gel pictures. Experiments were repeated three independent times with similar results. For (**h**) and (**l**), error bars show the mean ± SD from $n = 3$ technical replicates. Experiments were repeated twice with similar results. For (**j**) and (**n**), data are pooled from two independent experiments ($n = 4$) and error bars represent the mean ± SD. For (**o**), data are pooled from three independent experiments ($n = 3$) and error bars represent the mean ± SD. *P*-values were determined by unpaired two-tailed *t*-tests.

regulatory networks between RBPs and elucidates the role of STRAP in RNA biogenesis.

## Methods

**Generation of *Strap* +/− mice.** Mouse mutants with deletions were created using the previously described zinc finger nucleases (ZFNs) protocol[18]. In brief, ZFNs targeting the *Strap* gene were electroporated into V6.5 ESCs. *Strap*[+/−] or *Strap*[flox/flox] ESCs were then injected into C57BL/6 blastocysts to generate chimeric mice. The chimeras were backcrossed with C57BL/6 mice (10–12 week-old male and female, Jackson Laboratory). *Strap*[+/−] or *Strap*[flox/flox] offspring were subsequently intercrossed for the indicated experiments. Postimplantation embryos were isolated from naturally mated timed-pregnant mice. To genotype embryos, yolk sacs were lysed at 55 °C overnight in lysis buffer (10 mM Tris-HCl pH 8.3, 50 mM KCl, 2 mM MgCl2, 0.45% NP40, and 0.45% Tween-20) supplemented with 1 μg/ml proteinase K (Invitrogen). The primers used for genotyping are listed in Supplementary Data 10. All animal experiments were carried out according to the guidelines for the care and use of laboratory animals of the University of Alabama at Birmingham Institutional Animal Care and Use Committee (IACUC). We have complied with all relevant ethical regulation for mice testing and research. Animals were housed at a controlled temperature (23 °C) and humidity (55%) under a 12:12 h light-dark cycle and received standard mouse chow and water.

**Derivation of mouse ESC lines.** E3.5 blastocysts were collected from *Strap*[+/−] females mated with *Strap*[+/−] males. ESC lines were derived from E3.5 blastocysts in defined medium (Knockout D-MEM medium, 20% Knockout Serum Replacement, penicillin/streptomycin, 2 mM L-glutamine, 1× MEM non-essential amino acids, 100 μM ß-mercaptoethanol, 1000 U/mL recombinant mouse LIF) and cultured on MEF-feeder cells. We generated a total of 24 independent cell lines: 5 wild type, 14 heterozygous, and 5 *Strap*[−/−].

**ESCs and EBs culture.** All established ESC lines were routinely cultured on MEF-feeder cells in standard ESC culture medium (Knockout D-MEM medium, 15% ES-qualified FBS, penicillin/streptomycin, 2 mM L-glutamine, 1 mM sodium pyruvate, 1× MEM non-essential amino acids, 100 μM ß-mercaptoethanol, and 1000 U/mL recombinant mouse LIF). Mouse E14 ESCs were cultured on feeder-free condition[18]. The medium was changed daily. Every other day cells were passaged by trituration of the colonies after trypsinization.

EBs were formed in EB medium (ESC medium without mLIF) in hanging drops for 48 h and then cultured in 6 well ultralow attachment plates for an additional 48 h. For neuroectoderm differentiation, EBs were cultured in N2B27 medium (1:1 mixture of DMEM/F-12 medium and Neurobasal Medium, which was supplemented with basic fibroblast growth factor (bFGF; 10 ng/mL), N2 supplement, B27 supplement, and 1× penicillin/streptomycin/glutamine) for 4 days. EBs further underwent neuronal differentiation for additional days in N2B27 medium (1:1 mixture of DMEM/F-12 medium and Neurobasal Medium supplemented with N2 and B27 supplement, 1× penicillin/streptomycin/glutamine and ascorbic acid (200 μM))[32].

**Whole-mount in situ hybridization.** Embryos were fixed overnight at 4 °C in 4% paraformaldehyde (PFA) in PBS, dehydrated in increasing methanol concentrations in PBST (PBS + 0.1% Tween-20), and stored in 100% methanol at −20 °C until further processing, according to a standard protocol with minor modifications. Briefly, to detect the hybridized signal, embryos were washed with MBST (100 mM maleic acid, 150 mM NaCl, containing 0.1% Tween-20), pre-blocked in 10% sheep serum, 2% Boehringer Blocking Reagent (Roche) at room temperature for 90 min and then incubated overnight at 4 °C with a 1:2000 dilution of an alkaline phosphatase (AP)-conjugated anti-digoxigenin antibody (Roche) in blocking solution. After several washes in MBST of 1 h each, an additional overnight wash at 4 °C was performed. Detection of staining was performed using the

BM purple AP substrate (Roche) for 8 h. The embryos were post-fixed with 4% PFA overnight at 4 °C and stored in 70% ethanol. Sense (negative-control) and antisense RNA probes for *Strap* were generated by PCR. Digoxigenin labeling was performed using the DIG RNA Labeling kit and T3 RNA polymerase (Roche). Primer sequences are listed in Supplementary Data 10.

**Teratoma formation assay.** $5 \times 10^6$ cells were injected subcutaneously into the flanks of female NON-SCID mice. After 4–5 weeks, teratomas were isolated, transferred into Bouin's fixative overnight and subjected to immunohistological examination with the following antibodies: NESTIN (at a dilution of 1:150, Cell Signaling, #4760); SMA (at a dilution of 1:70, R&D systems, MAB1420-SP); or AFP3 (at a dilution of 1:200, R&D systems, MAB1368-SP).

**Nuclear extracts (NEs) preparation.** Mouse ESCs ($1 \times 10^9$) were homogenized in Buffer A (10 mM Tris-HCl (pH 7.9), 10 mM KCl, 10% glycerol, 1.5 mM MgCl2) supplemented with fresh 0.5 mM DTT, protease inhibitors (Roche), and 0.2 mM PMSF) on ice. Nuclei were sedimented by centrifugation ($1000 \times g$, at 4 °C) and suspended in Buffer C (20 mM Tris-HCl (pH 7.9), 1.5 mM MgCl2, 0.42 M NaCl, 0.2 mM EDTA, 25% glycerol, 0.5 mM DTT and protease inhibitors). Soluble nuclear proteins were separated by centrifugation ($25,000 \times g$ for 30 mins, at 4 °C) and dialyzed against BC50 buffer (20 mM Tris-HCl (pH 7.9), 50 mM KCl, 0.2 mM EDTA, 10% glycerol, 10 mM ß-mercaptoehtanol) for 7 h. Samples were centrifuged at $16,000 \times g$ for 30 mins at 4 °C and the suspernatant was kept at −80 °C.

**Immunoprecipitation.** Mouse ESC NEs were treated either with RNase A (100 μg/mL) for 15 min or with DNase I (10 U) for 30 min at 37 °C. Immunoprecipitations were then performed with samples (with or without treatment) resuspended in standard immunoprecipitation buffer (50 mM Tris-HCl (pH 7.5), 150 mM NaCl, 10 mM EDTA, 0.02% NaN3, 50 mMNaF) and rotated overnight with beads cross-linked anti-STRAP antibody (4 μg/mg lysates, Bethyl, A304-735A). Rabbit IgG (3 μg/mg lysates, Cell signaling, #2729) was used as a negative control. For NE samples from mouse ECSs or EBs, immnoprecipitations were performed using either beads cross-linked with anti-SF3B1 antibody (4 μg/mg lysates Bethyl, A300-996A) or anti-CHERP antibody (at the dilution of 1:500, Santa Cruz, sa-100650).

**Proteomics analysis.** We independently performed the proteomics experiments twice with similar results. On each time, we had an NE-specific sample pulled-down by an anti-STRAP antibody ($n = 1$) and a parallel control sample precipitated by an anti-IgG antibody ($n = 1$). Complex proteins from each sample were separated by SDS Bis-Tris gel (4–12%, Invitrogen). The gels were stained with SyproRuby. The entire lane for each sample was partitioned into 6 MW fractions, and each gel plug was equilibrated in 100 mM ammonium bicarbonate (AmBc). Each gel plug was then digested with Trypsin Gold (Promega) following the manufacturer's instruction, and peptide extracts were reconstituted in 0.1% formic acid/ ddH2O at ~0.1 μg/μL. Peptide digests were injected onto a 1260 Infinity nHPLC stack (Agilent Technologies), and separated using a 75 micron I.D. × 15 cm pulled tip C-18 column (Jupiter C-18 300 Å, 5 micron, Phenomenex). This system ran in-line with a Thermo Orbitrap Velos Pro hybrid mass spectrometer equipped with a nano-electrospray source (Thermo Fisher Scientific), and all data were collected in CID mode. The nHPLC was configured with binary mobile phases that included solvent A (0.1%FA in ddH2O), and solvent B (0.1%FA in 15% ddH2O / 85% ACN), programmed as follows; 10 min @ 5%B (2 μL/min, load), 90 min @ 5%–40%B (linear: 0.5 nL/min, analyze), 5 min @ 70%B (2 μL/min, wash), 10 min @ 5%B (2 μL/min, equilibrate). Following each parent ion scan (300-1200 *m/z* @ 60k resolution), fragmentation data (MS2) were collected on the top most intense 15 ions. For data dependent scans, charge state screening and dynamic exclusion were enabled with a repeat count of 2, repeat duration of 30 s, and exclusion duration of 90 s.

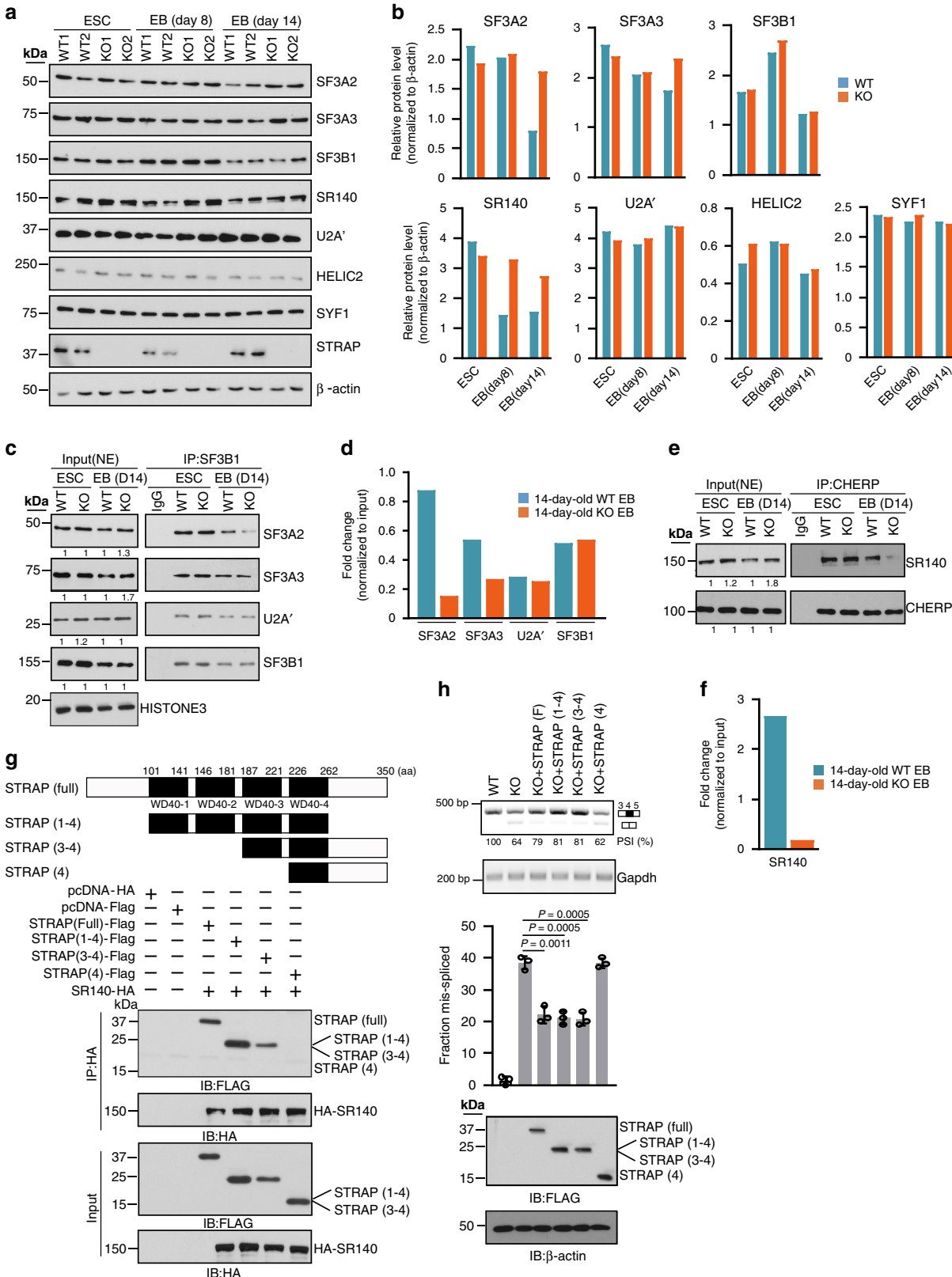

The XCalibur RAW files were further collected in profile mode, centroided and converted to MzXML using ReAdW (version 3.5.1). The mgf files were then created using MzXML2Search (included in TPP v. 3.5) for all scans. The data was searched using SEQUEST (v.27 rec12. dta files), which was set for two maximum missed cleavages, a precursor mass window of 20 ppm, trypsin digestion, variable modification C @ 57.0293, and M @ 15.9949. Searches were performed with a species specific subset of the UniRef100 database. The list of peptide IDs generated based on SEQUEST search results were filtered using Scaffold (version 3.0). The filter cut-off values were set with a minimum peptide length of >5 AA's, with no MH + 1 charge states, with peptide probabilities of >80% C.I., and with the number of peptides per protein ≥2. The protein probabilities were then set to a > 99.0% C.I., and an FDR < 1.0%. STRAP binding-partners were ranked by a COMPLEAT online tool (http://www.flyrnai.org>compleat)[29]. The enrichment P-value was calculated based on the permutation: generate 1000 random lists of input data then

**Fig. 7 Loss of STRAP has negative effects on 17S U2 snRNP biogenesis. a, b** Representative Western blots (**a**) and protein-level quantifications (**b**) of several spliceosomal components in WT and STRAP KO cells at the indicated days. All data were normalized to ß-Actin. Data are shown from one of two independent experiments; bars represent as means (n = 2, biological replicates). **c–f** Representative Western blots (**c** and **e**) and co-immunoprecipitated protein-level quantifications (**d** and **f**) showing co-IP results. All data were normalized to the respective input. The expression levels of indicated proteins (relative to its parallel WT controls) are shown below the gel pictures. Data, pooled from two independent experiments; bars represent as means. **g** Schematic illustration of STRAP-truncated mutants (upper panel) and co-IP experiments to map the domain(s) of STRAP required for binding of SR140 (bottom panel). 293T cells were transfected either with empty vectors or expression vectors as indicated. Cellular extracts were then subjected to immunoprecipitation with an HA antibody followed by Western blot analysis using a Flag or HA antibody. Expressions of transfected proteins were also determined. The experiment was repeated three independent times with similar results. **h** Gain-of-function assay using UPF3A mini-gene in MEF WT and *Strap* KO cells. Indicated cells were co-transfected with miniUPF3A and an empty vector or a vector encoding various Flag-tagged STRAP, as shown in **g**. RT-PCR assays were performed to detect alternative exon inclusion in miniUPF3A. Quantifications of PSI values are shown below the gels, and the fractions of mis-splicing are pooled from three independent analyses (n = 3). Error bars represent the mean ± SD. The Flag-tagged protein levels and ß-Actin were assayed by Western blot analysis. ß-Actin was used as a loading control. *P*-values were determined by unpaired two-tailed *t*-tests.

calculate the *P*-value based on the distribution of the scores with random lists[29]. *Sub*-networks for STRAP interaction with spliceosomal partners were also computed by this tool.

**Immunofluoresecence staining**. Cells were fixed in 4% paraformaldehyde for 10 min at room temperature, blocked, and then incubated at 4 °C overnight with the following antibodies: SR140 (at the dilution of 1:150, Santa Cruz, sc-398718); SF3A2 (at the dilution of 1:150, Santa Cruz, sc-390444); SF3B1 (at the dilution of 1:150, Santa Cruz, sc-514655); SYF1 (at the dilution of 1:150, Santa Cruz, sc-271037); or STRAP (at the dilution of 1:200, Bethyl, A304-735A). After washing, cells were incubated with goat anti-rabbit Alexa Fluor 488 antibody (at the dilution of 1:150, Life Technologies, A-11008) or goat anti-mouse Alexa Fluor 555 antibody (at the dilution of 1:150, Life Technologies, A-21422) and counter-stained with DAPI to detect nuclei.

**Western blotting**. Proteins were separated by 10% SDS-PAGE and probed with primary antibodies. Primary antibodies, at the dilution of 1:1000 for each, included: SNRPA (Santa Cruz, sc-376027); U2A' (Santa Cruz, sc-393804); SR140 (Santa Cruz, sc-398718); HELIC2 (Santa Cruz, sc-393170); PRPF3 (Santa Cruz, sc-101130); SYF1 (Santa Cruz, sc-271037); SF3A3 (Santa Cruz, sc-393673); SF3A2 (Santa Cruz, sc-390444); SF3B1 (Santa Cruz, sc-514655); STRAP (BD Transduction Labs, #611346); DDX15 (Santa Cruz, sc-271686); TFIIIC110 (Santa Cruz, sc-81406); GAPDH (Cell Signaling, #2118), and CHERP (Santa Cruz, sc-100650). β-Actin (Sigma, A5316) was used at the dilution of 1:10,000.

**Glutathione S-transferase (GST)-fused protein purification and RNA EMSA**. Full-length STRAP was PCR amplified from previously reported pcDNA3-STRAP-Flag vector[60] and subcloned into pGEX-4T-1 GST expression vector using BamHI and XhoI restriction sites. The induction was performed by adding 0.4 mM isopropyl-β-D-thiogalactopyranoside at 30 °C for 3.5 h. Whole bacterial lysates were applied to a glutathione Sepharose 4B (GE Healthcare Life Science, 45000139) column, and GST-tagged proteins were purified according to the manufacturer's instructions. To perform RNA-EMSA and supershift analyses, a Chemiluminescent RNA EMSA kit (Thermo Fisher Scientific, #20158) was used. Briefly, binding reaction mixtures (10 µL) contained 10× binding buffer, 2 µg tRNA, 5% (vol/vol) glycerol, 20 mM DTT, 2 nM biotin-labeled RNA oligonucleotides and various concentrations of GST-proteins. After incubation for 30 min at room temperature, the samples were separated on 6% polyacrylamide gels followed by detection with an enhanced chemiluminescence system.

**RNA immunoprecipitation (RIP) assay**. Cells were irradiated with ultraviolet light at 254 nm (400 mJ/cm$^2$). Cells were then harvested and lysed in 2 ml assay buffer (30 mM Tris-HCl pH 7.6, 0.5% Triton X-100, 2.5 mM MgCl$_2$, 100 mM NaCl, 1 mM DTT, 80 U/ml RNase OUT (Invitrogen))[61]. After centrifugation, the clear lysates were immunoprecipitated by an anti-STRAP antibody (5 µg/mg lysates, Bethyl, A304-735A) at 4 °C overnight. Beads were washed with lysis buffer, treated with 20 units of RNase-free DNase (Roche) for 15 min at room temperature, and treated with 50 µg of proteinase K (Roche) for 30 min at 37 °C. RNAs were isolated from the supernatant using Trizol (ThermoFisher Scientific).

**RT-PCR and qPCR**. To determine gene expression or splicing efficiency, total RNAs were extracted with Trizol and reverse-transcribed using the SuperScript III First-Strand Synthesis System (Invitrogen) with random primers. For gene expression and RIP assays, qPCR was performed with 2× SYBR-Green Master Mix (Roche) on Light Cycler 480 (Roche) machine. The cycling acquisition program was as follows: 95 °C 10 min, 40 cycles of 95 °C for 10 s, 55 °C for 20 s, 72 °C for 20 s, and 60 °C for 5 s. For detection of splicing, PCR was performed using GoTaq Green Master Mix (Promega) with 95 °C for 2 min, then 30 cycles of 95 °C for 20 s, 52 °C 30 s, and 72 °C 50 s. PCR products were resolved on agarose gels, and the

signal intensities of the bands were quantified by the ImageJ program. Primers used are listed in Supplementary Data 10.

**Expression plasmids**. The truncated coding regions from the full-length STRAP-Flag vector[60] were inserted into the pcDNA3.0 Flag vector using BamHI and XhoI restriction sites. The primers used for subcloning are listed in Supplementary Table 10. Plasmid SR140-HA was a gift from Dr. Ying Feng (Chinese Academy of Sciences, China)[49]. For co-IP assays, HEK293T cells were transfected with various plasmids using Lipofectamine3000 (Thermo Fisher Scientific), and lysates were subjected to co-immunoprecipitation with an anti-HA antibody (3 µg/mg lysates, Bethyl, A190-208A) and probed with either an antibody for anti-Flag (at the dilution of 1:15,000, Sigma, F3165) or anti-HA (at the dilution of 1:1000, Bethyl, A190-208A).

**Mini-gene reporter assays**. *Nnat* and *Ppp2r2d* mini-genes were constructed by amplifying genomic sequences spanning exons 1–3 of *Nnat* and exons 3–5 of *Ppp2r2d* respectively, which were then cloned into pcDNA3.1 vectors (BamHI and XhoI restriction sites for *Nnat*, KpnI, and EcoRI restriction sites for *Ppp2r2d*). The primers used for subcloning are listed in Supplementary Table 10. The *UPF3A* mini-gene with exons 3–5 was kindly provided by Dr. Ying Feng[49]. Mini-gene vector and STRAP truncating variants were co-transfected into MEF cell lines using Lipofectamine3000 reagent and the cells were cultured for a further 48 h until RNA isolation.

**Inducible STRAP knockdown in mouse ESCs**. For RNA interference in mouse E14 ESCs, an shRNA against *Strap* was cloned into pLKO-Tet-On (AgeI and EcoRI sites). The target sequences of shRNA was as follows: GGCAGGGA-TATTCACCATTATCGTTTCAGA. For the knockdown experiment, pLKO-Tet-On-based lentiviral vectors and packaging plasmids pMD2.G and psPAX2 were co-transfected into 293T cells using Lipofectamine 3000 reagent. The supernatant was collected after 48 h and passed through a 0.45 µm filters (Millipore). E14 cells were cultured in the viral supernatant in the presence of 5 µg/mL polybrene (Sigma) for 48 h. After infection of cells, selection was started by adding 4 µg/mL of puromycin to E14 ESCs. ESCs were then formed to EBs and underwent differentiation in N2B27 medium. shSTRAP expression was induced by the addition 1 µg/mL of Dox to the culture medium of 11-day-old EBs.

**Patch clamp recordings and flow cytometry**. Whole cell patch-clamp recordings in current-clamp mode were performed with a borosilicate microelectrode filled with a potassium gluconate-based internal solution. EBs were held at resting membrane potential (RMP) and 500 ms current steps from −200 pA to 700 pA were delivered. Single cells digested from EBs were collected at time points and stained with CD24-FITC (2 µL/test, MACS, #130-110-825) and CD56-APC (2 µL/test, R&D systems, FAB7820A) antibodies followed by flow cytometry. Becton Dickinson FACSDiva (version 8.0) analyzed the positive population.

**RNA-seq library preparation and sequencing**. For analysis of differential gene expression, samples were lysed with Trizol, treated with DNase (QIAGEN) and purified using RNeasy Minielute Cleanup Kit (QIAGEN). Ribosomal RNA was removed from each RNA extraction using Ribo-Zero Gold rRNA Removal kit (Illumina). Indexed libraries were pooled and sequenced at a final concentration of 1.8 pmol/L on an Illumina NextSeq 500 using paired-end chemistry with a 75-bp read length. Cutadapt (version 2.2) was used for trimming primer adapters from raw FASTQ files. Sequencing reads were mapped to Gencode GRCm38 p4 Release M11 using STAR version 2.5.2b (options:–outReadsUnmapped Fastx;–outSAMtype BAM SortedByCoordinate;–outSAMattributes All). Transcript abundances were calculated using Cufflinks version 2.2.1 with options–library-type fr-firststrand; -G; -L. Cuffmerge was then used to merge the transcript files from Cufflinks into one file. Following Cuffmerge, Cuffquant was used to quantify the

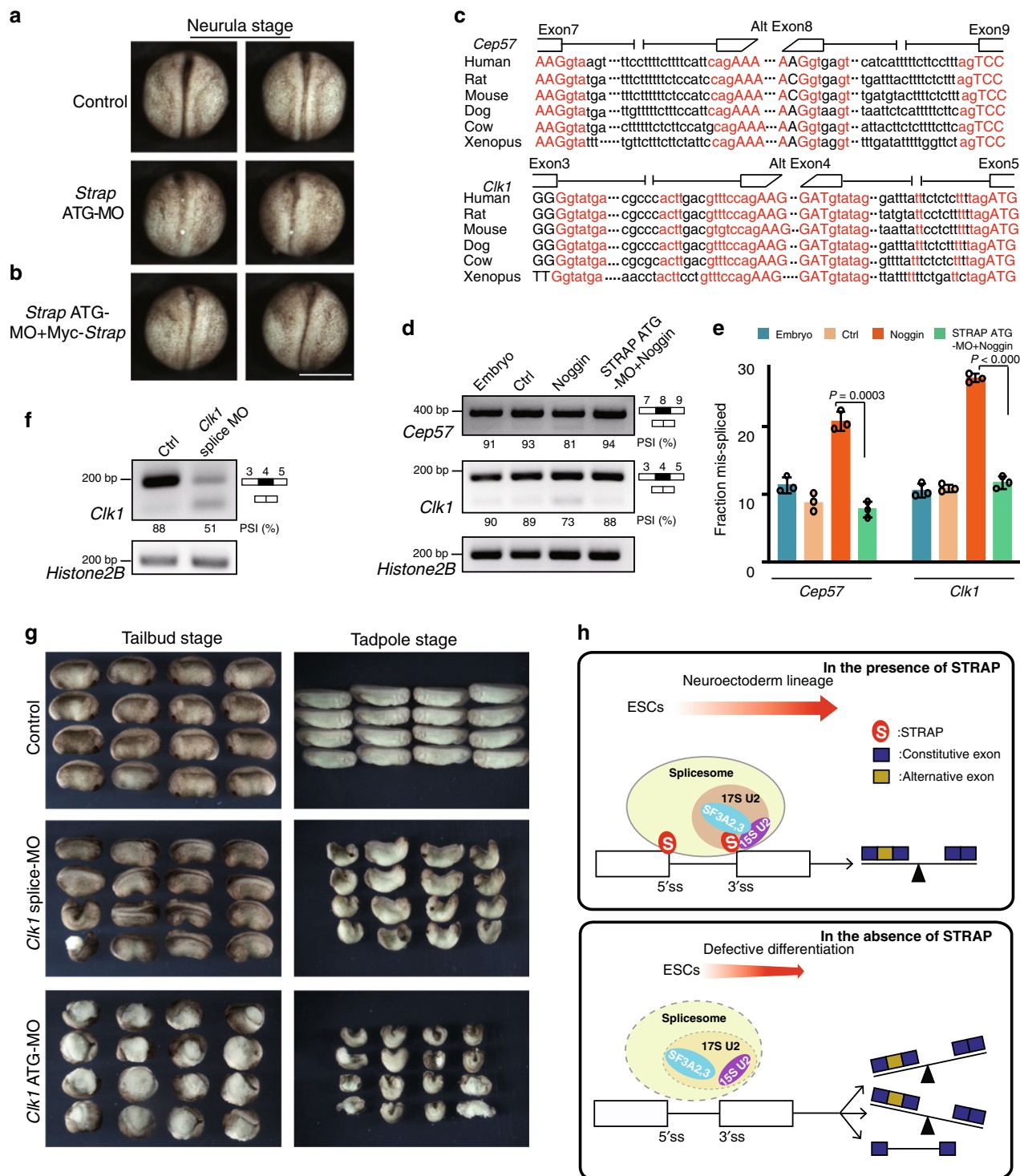

transcript abundances, followed by differential gene expression using Cuffdiff. Differentially expressed genes with a *P* value <0.01, as well as a log$_2$ fold change >1 were further analyzed further.

**Splicing analysis from RNA-seq data**. AS was inferred from the RNA-seq data using the Multivariate Analysis of Transcript Splicing (rMATS) version 3.2.5 software for replicates, which processed the above STAR alignments (in BAM format) to provide the AS events[17]. Parameter values for rMATS were set by optimizing the accuracy of the MATS estimation of splicing difference of known standards (genes previously analyzed by RT-PCR). The final rMATS analysis of STRAP-responsive alternative splicing was accomplished with options -b1, -b2, -gtf, -t paired, -len 74, and -novelSS1. The output from rMATS was then filtered using the criteria of: FDR < 0.05 and absolute value of ΔPSI ≥ 10% (unless otherwise noted). MAJIQ and Voila (https://biociphers.bitbucket.io/majiq/index.html) were used to detect[37],

quantify and visualize local splicing variations (LSVs) from the RNA-Seq data. Briefly, the MAJIQ build tool used the alignment BAM files from STAR along with a gene annotation file (GFF3) to define splice graphs and known/novel LSV. Following the build, MAJIQ psi and deltapsi tools were used to quantify the relative abundances (PSIs) of LSVs and changes in relative LSV abundance (delta PSI) between the two conditions. The Voila tsv tool was used to create a tab-delimited text file (–threshold 0.1) in order parse the MAJIQ results and further analyze particular LSVs or genes of interest.

**5′ and 3′ splice sites strength**. To compute a score for the 5′ and 3′ splice sites, we used the maximum entropy models for splice sites (MaxEntScan). The 5′ss scoring uses a 9-bases long (the last 3 bases of the exon and the first 6 bases of the downstream intron), while the 3′ss scoring uses a 23-bases long (the last 20 bases of the upstream intron and the first 3 bases of the exon).

**Fig. 8 Loss of STRAP triggers aberrant AS across species. a, b** *Strap*-MO (50 ng) was injected alone or with Myc-*Strap* (1 ng) into the marginal zone region of two dorsal blastomeres in 4-cell-stage *Xenopus* embryos. The embryos were cultured to the neurula stage; representative embryos are shown. The experiment was repeated three independent times with similar results. MO, morpholino oligonucleotides. Scale bar, 1 mm. **c** Evolutionary conservation of sequence (red) across vertebrates within the putative splicing regions in the *Cep57* and *Clk1* genes. Extracted splicing sequences are referred to by MaxEntScan definition. **d** RT-PCR analysis of splicing in the presence of Noggin RNA with or without *Strap* antisense MO in *Xenopus* early tailbud stage. Quantifications of PSI values are shown below the gel. **e** Fraction of mis-splicing calculated from one of three independent analyses is shown in the bar graph. Error bars indicate the mean ± SD from $n = 3$ biological replicates. $P$-values were determined by unpaired two-tailed $t$-tests. **f** *Clk1* splice-MO (50 ng) was injected into the dorsal animal regions of 4-cell to 8-cell stage *Xenopus* embryos. RT-PCR analysis of splicing with *Clk1* splice-MO in *Xenopus* at the early tailbud stage. Quantifications of the PSI values are shown below the gels. The experiment was repeated three independent times with similar results. **g** Embryos were cultured until the tailbud or tadpole stages; representative embryos are shown here. The experiment was repeated three independent times with similar results. **h** Model describing the role of STRAP in mediating AS patterns. Upper, under conditions of normal levels of STRAP, it cooperates with the 17S U2 snRNP complex to regulate certain splicing sites, resulting in a balance of altered splicing patterns during neuroectoderm lineage commitment. Bottom, loss of STRAP has a negative effect on assembly of the 17S U2 complex. The fidelity of pre-mRNA splicing is also disrupted, causing high or low alternative transcripts or retaining ones. Blue box: constitutive exon; Yellow box: alternative exon; S: STRAP; 17S U2: 17S U2 snRNP complex; 15S U2, 15S U2 snRNP complex; 5′/3′ ss: 5′/3′ splicing site. The schematic diagram was created by the authors.

**STRAP-RNA binding assay**. Nucleotide resolution UV crosslinking at 254 nm (400 mJ/ cm²) and immunoprecipitation were performed. Two replicates from two independent WT EB or embryos were used for generating protein-RNA binding samples. Lysates generated from the crosslinked cells were treated with Turbo DNase (Ambion) and RNase I (1: 50 for high or 1:500 for low dilution, Ambion) for 5 min at 37 °C to digest the genomic DNA and trim the RNA to short fragments of an optimal size range. RNA-protein complexes were immunoprecipitated with 100 μl of protein A Dynabeads (Life Technologies) and 10 μg of anti-STRAP (Bethyl, A304-735A) antibody. Following stringent high salt washes (50 mM Tris-HCl; 1 M NaCl; 1 mM EDTA; 1% NP-40; 0.1% SDS; 0.5% sodium deoxycholate), the immunoprecipitated RNA was 5′ end-labeled with radioactive ³²P γ-ATP[62]. The immunoprecipitated complexes were separated by SDS-PAGE and transferred to a nitrocellulose membranes followed by the exposure to a Fuji film at −80 °C.

**Establishment of eCLIP-seq libraries**. eCLIP libraries were generated based on the standardized eCLIP experimental protocol[38] with minor modifications. Briefly, UV-crosslinked WT EBs (254 nm; 400 mJ/cm²) or embryos were lysed in iCLIP lysis buffer[62] and sonicated (BioRuptor). Lysates were treated with RNASE I (Ambion) to shear RNA, after which STRAP (Bethyl, A304-735A), and rabbit IgG (Cell signaling, #2729) protein-RNA complexes were immunoprecipitated using the indicated antibody. In addition to the RBP-IPs, a parallel size-matched input library was generated. One input library was used for each biological replicate eCLIP group. Stringent washes were performed as described in iCLIP[62], during which RNA was dephosphorylated with FastAP (Fermentas) and T4 PNK (NEB). Subsequently, a 3′ RNA linker was ligated onto the RNA with T4 RNA ligase (NEB). Protein-RNA complexes were run on SDS-PAGE gels and transferred to nitrocellulose membranes, and RNA was isolated from the membranes according to standard iCLIP procedure. After precipitation, RNA was reverse transcribed with AffinityScript (Agilent) followed by cDNA clean-up (ExoSap-IT, Affymetrix), and a 3′-DNA linker was ligated onto the cDNA product with T4 RNA ligase (NEB). Libraries were then amplified with Accuprime Supermix I (Invitrogen).

**eCLIP-seq analysis**. To perform this analysis, we followed the eCLIP-seq pipeline[38] with minor modifications. Sequencing reads were first trimmed using Cutadapt (version 2.2) with options -f fastq–match-read-wildcards–times 1 -e 0.1 -O 1–quality-cutoff 6 -m 18 -a NNNNNAGATCGGAAGAGCACACGTCTGAACTCCAGTCAC -g CTTCCGATCTACAAGTT -g CTTCCGATCTTGGTCCT. The trimmed sequences were then mapped to the mouse RepBase to remove repetitive elements using STAR version 2.5.2b with options:–outSAMunmapped Within–outFilterMultimapNmax 30–outFilterMultimapScoreRange 1–outSAMattributes All–outStd BAM_Unsorted–outSAMtype BAM Unsorted–outFilterType BySJout–outReadsUnmapped Fastx–outFilterScoreMin 10–outSAMattrRGline $RG–alignEndsType EndToEnd. Then, the unmapped reads from the above STAR alignment were mapped to Gencode GRCm38 p4 Release M11 using STAR with options:–outSAMunmapped Within–outFilterMultimapNmax 30–outFilterMultimapScoreRange 1–outSAMattributes All–outStd BAM_Unsorted–outSAMtype BAM Unsorted–outFilterType BySJout–outReadsUnmapped Fastx–outFilterScoreMin 10–alignEndsType End-ToEnd. Peaks were then called on the reads that mapped to the mouse genome using MACS version 1.4.2 with options: -t -c–name–format = "BAM" -g mm–tsize=75. Homer was used to annotate the peaks. In total, six eCLIP libraries (including INPUTs) were sequenced to 2.2–39.7 million reads, which were mapped uniquely to the mouse genome (mm10) allowing 0–2 mismatches. Using the stringent enrichment criteria ($p \leq 10^{-3}$ and >5 fold-enriched versus the respective Input), a transcriptome-wide set of 13,952 and 14,042 high-confidence peaks in EBs and 16,833 and 11,909 in embryos were identified, respectively. Tracks were visualized using the Integrative Genomics Viewer (IGV).

**eCLIP correlation analysis**. To evaluate the biological reproducibility of eCLIP data, pairwise comparisons were performed to use $1.4–2.3 \times 10^5$ uniquely mapped reads without PCR duplicates. All reads with read depths >3 were obtained using Samtools (version 1.3.1) and in-house Perl (version 5.26.3) scripts. The resulting $\log_2$ (Read depth) values were then plotted with the ggplot2 (version 3.2.1) package in R, followed with calculation of the Pearson correlation coefficient.

**Merged eCLIP-seq peaks**. As the resulting eCLIP reads were highly correlated, we merged all eCLIP clusters to ensure both sensitivity and specificity across the reproducible data[63]. The merged peaks were generated based on the following strategy: (1) all eCLIP clusters common to, at least two of four libraries, would be chosen; and (2) if the two peaks from various groups have intersected regions, both of them would be retained and combined. After merged peaks were generated, the self-comparison was performed, and the redundancy was eliminated. In-house Perl (version 5.26.3) scripts were used for comparison of peak regions and generation of the final list of merged peaks.

**Motif analysis**. Motif analysis on peak regions was performed using HOMER software (http://homer.ucsd.edu/homer/ngs/peakMotifs.html)[64], with a combined list of peaks as the input. The parameters for the main script "findMotifsGenome.pl" in HOMER software were: findMotifsGenome.pl <peak/BED file > <genome > <output directory > -size given –rna –mask. In the result file, the $P$-value was used to rank the enrichment level of motifs.

**RNA map analysis**. Based on genomic coordinates of each peak, the gene and transcript information was obtained through Mutalyzer (version 2.0.29 https://mutalyzer.nl) and the corresponding read depth information was extracted from the BAM files using Samtools (version 1.3.1). If one peak region covered multiple genes/transcripts, generally, the longest protein-coding gene/transcript would be chosen for following analysis. In exon-intron/intron-exon boundary analysis, two kinds of boundaries were included in the analysis: (1) the boundaries that were covered by at least one peak; and (2) the exon-intron or intron-exon boundaries that were closest to one of two terminals of a peak, but not covered by this peak. For example, if peak-1 covered a part of exon-3, intron-3, exon-4, and a part of intron-4, then the boundaries between (1) intron-2 and exon-3, (2) exon-3 and intron-3, (3) intron-3 and exon-4, (4) exon-4 and intron-4, (5) intron-4 and exon-5 would be included in the analysis. This resulted in 11456-19058 unique exon–intron boundaries and 11444-16993 intron–exon boundaries. In-house Perl (version 5.26.3) scripts were created for selection of genes/transcripts and to count the number of CLIP sequences covering each base for a distance of 100 bases in the exon and 300 bases in the intron from all exon boundaries. For each position of a boundary, the relative density was defined as the total number of reads in the sample/the total number of reads in the control. For composite maps for STRAP-binding peaks on induced exon-inclusion or—skipping events, a window between two constitutive exons was taken and divided into 100-bp equally sized bins. Only normalized peak size by allowing a given position was considered as an occupation.

**Gene ontology analysis**. Gene Ontology (GO) enrichment analysis was performed with in-house Perl scripts based on gene-GO association file (ftp://ftp.geneontology.org/go/gene-associations/gene_association.mgi.gz), and $P$-values were determined by Chi-squared test followed by Bonferroni correction.

**Xenopus**. *Xenopus* laevis frogs (1–2 years old male and female, Nasco) were used following the institutional IACUC protocol 09658 at the University of Alabama at Birmingham. We have complied with all relevant ethical regualtions for animal testing and research. Female frogs were primed with 800 units/frog of human chorionic gonadotropin hormone (Sigma-Aldrich) the night before use[65]. Embryos were

obtained by in vitro fertilization and dejellied with 2% cysteine solution before injection of indicated molecules. The translational-blocking antisense morpholino oligonucleotide (MO) was designed using Gene Tools service as follows: 5′-CCAC TAGCGAGGGCTTCATGTCAAT-3′ for *Strap* targeting the translation start site (ATG-MO); 5′-CTTCGGTTGCTGTGGTTCATCTGTT-3′ for *Clk1* ATG-MO; and 5′-TTCGATGACTCTTCTGGAAACAGGA-3′ for *Clk1* spliced isoform (splice-MO). We performed PCR-based cloning to add the Myc-tag at the N-terminus of *Strap* or *Clk1*. The PCR products were cut with NotI/XhoI for *Strap* and BamHI/XhoI for *Clk1* and inserted into a pCS105 vector. *Strap* ATG-MO (50 ng), with or without *Strap* RNA (1 ng), was injected into the marginal region of two dorsal blastomeres in 4-cell-stage embryos ($n = 12$–$16$ per group) and collected at the neurula stage for phenotypic analysis. Noggin RNA (10 pg) was injected with or without *Strap* ATG-MO (50 ng) into the animal regions of both blastomeres in 2-cell stage embryos ($n = 12$–$16$ per group). The animal caps were dissected at late blastula stage 9 and cultured until the tailbud stage before RNA was extracted. *Clk1* ATG-MO (50 ng) or splice-MO, with or without *Clk1* RNA (0.5 ng), was injected into the dorsal animal regions of 4-cell to 8-cell stage embryos ($n = 12$–$16$ per group). The embryos were cultured until the tailbud or tadpole stages. The morphology of the embryos was recorded using a Nikon AZ100 microscope.

**Statistics and reproducibility**. Unless otherwise stated, multiple independent experiments were performed to verify the reproducibility of all experimental findings, unless otherwise stated. All quantitative data represent the means ± sd. Significance tests were unpaired two-tailed Student's *t*-tests, One-way ANOVA tests, two-sided unpaired Wilcoxon tests, Chi-squared tests or Fisher's extract tests. The Spearman correlation coefficients were calculated based on the ΔPSI values between our data and the previously reported datasets. Pearson's correlation coefficient was used to evaluate the pairwise correlation between groups. All of the statistical analysis were performed using R package (version 3.3.3) or Prism version 7.0 (GraphPad).

**Reporting summary**. Further information on research design is available in the Nature Research Reporting Summary linked to this article.

## Data availability

The mass spectrometry proteomics data have been deposited in the ProteomeXchange via the PRIDE database with identifier PXD015371. RNA-sequencing and eCLIP-sequencing data that support the findings of this study have been deposited in the Gene Expression Omnibus under accession codes GSE131474. Hallmark genes signatures used in Supplementary Fig. 4e, f were obtained from the publicly available sources. Among them, gene expression data for neuroectoderm-like EBs and mouse E14.5 brain were from Gene Expression Omnibus database under the accession code GSE116153 and GSE30765, respectively. For mouse E8.25 tissue, gene expression data were obtained from European Nucleotide Archive under the accession code PRJEB4513. All the other relevant data supporting the findings of this study are available within the article and its Supplementary Information files or from the corresponding author upon the reasonable request. The source data underlying Supplementary Figs 2b–d, 3b, 3d, e, 4b–d, 4k, 6a–l, 7a–d, and 8a–h are provided as a Source Data file with this paper. Source data are provided with this paper.

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

## Acknowledgements

The authors thank the following individuals: E. Van Nostrand and J. König for assistance with eCLIP-seq and iCLIP-seq protocols, respectively; X.D. Fu and Y. Zhou for suggestions with eCLIP-seq bioinformatics analyses; D. Brooke for assistance with data analyses; D.D. Licatalosi, Y. Fondufe-Mittendorf, and H.Jiang for help with RNA-protein binding analyses; L.L. McMahon for initial help in electrophysiology studies; M. Athar for providing immunofluorescence microscopy; Y. Peng and K. Jiao for help with mouse embryo dissection; R. Serra for providing the WISH protocol; M. Crowley for sequencing; I. Popov for generating the expression plasmid for *Xenopus*; E. Ahn and R.S. Welner for discussing the results and reading the manuscript; and T. Smith and D. Hill for editing the manuscript. This work was supported by NIH R01CA95195, Veterans Affairs Merit Review Award (I01BX003497), a Faculty Development Award from UABCCC (P30 CA013148) and UAB U54 Pilot Project (CA 118948) (to P.K. Datta). It was also supported by NIH R01 130696-01A1 (to H.B. Wang) and NSF ISO-1558067 (to C.B. Chang).

## Author contributions

L.J. conceived the study, conducted the experiments, analyzed the data and wrote the manuscript; Y.C. and D.K.C performed bioinformatics analyses and data interpretation; A.D. assisted with *Strap* mice experiments; T.V. assisted in qPCR assays; J.A.M. performed mass spectrometry and analyzed data; M.K.B. helped with RNA-seq analysis; M.S. performed electrophysiology experiments; H.B.W. helped with proteomics experiments; C.B.C. performed *Xenopus* experiments. P.K.D. conceived, supervised the study, wrote the manuscript, and provided financial support. All authors read and approved the final manuscript.

## Competing interests

The authors declare no competing interests.

## Additional information

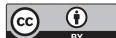

