## [Peer Review File · Nature Communications]

Reviewers' comments:

Reviewer #1 (Remarks to the Author):

This manuscript by Jin et al. shows STRAP knockout affects many alternative splicing events and early organogenesis during embryonic development. The authors first found Strap KO exhibits dysmorphogenesis as early as E8.5-9 and lethality at E10.5. Transcriptome analysis showed a major defect in alternative splicing regulation but modest transcriptional abnormalities. Proteomics on Strap co-IP samples identified spliceosome components as Strap's major interacting partners in the nucleus, suggesting STRAP be a splicing regulator. The authors used eCLIP to locate STRAP's RNA binding sites and found many in introns. Although the informatics data are largely descriptive, many are derived from *in vivo* samples, which is a significant strength. The manuscript is easy to read.

As presented, the novelty of the paper is proposing STRAP as an alternative splicing regulator. Some clarifications are needed (see below). The novelty can be enhanced with mechanistic studies of how STRAP regulates splicing and/or functional analysis of STRAP's targets.

Some specific comments (major):

1. Fig2: is the STRAP-spliceosome interaction RNA dependent? Because TFIIC complex is pulled down, the interaction could be even DNA dependent? The authors can answer these questions with DNase and RNase treatment of cell lysates. If the authors decide to pursue the mechanistic aspect, mapping of interaction domains and functional consequence of interaction mutations should be tested.
2. Does STRAP directly interact with RNA? CLIP is powerful but still has false positives and limitations. The author can use EMSA to measure STRAP binding to RNA with the CLIP motifs or any proposed STRAP gene targets (fig 4e).
3. RNA-seq targets and eCLIP-Seq targets do not substantially overlap (fig 5a). This does not necessarily invalidate the authors' conclusions, but the authors should provide a thorough explanation.
4. To show STRAP as a splicing regulator, the author can test whether removal or mutation of the STRAP binding motif would abrogate STRAP's regulation of a specific exon. The authors can pick a few to test from the 90 candidates (fig5a). These can be done in minigenes or cell lines.
5. The phenotypic characterization of the KO mice is rather limited and need to be elaborated.
6. Although ascribing a phenotype to specific splicing changes are challenging in this context, the authors need to show some STRAP-controlled splicing changes are functionally relevant to implicate the misregulated splicing underlies the phenotype. For example, do two isoforms have different biochemical functions? Does the splicing change impact cell physiology?
7. An important question is whether STRAP is a general splicing regulator or a lineage/stage specific regulator. STRAP's expression patterns and its KO/KD in a context other than germline deletion can answer this question. The latter also helps to mitigate the concern of indirect effects from germline deletion for 8 days.

Minor points:

1. Fig1a: please use fold changes or expression values in heatmap. Z-scores are influenced by variances and not really informative.
2. Fig1f: the scamp5 gel image is not consistent with the browser tracks.
3. Fig2g: what does each sub-network represent? What does a link between proteins mean? PPI?

4. Fig3h-i: Variability between the two biological replicates appears very high. how robust are these targets controlled by STRAP? For example, plekhg4 in fig3i does not appear to be regulated in strap KO.
5. Supp fig3d: looks like a large variance between biological replicates?

Reviewer #2 (Remarks to the Author):

This manuscript reports a study of STRAP as a novel regulator of alternative splicing in early mouse embryos. The authors carried out global studies of STRAP using RNA-seq, eCLIP-seq, complemented by experimental validations and individual assays, in knockout models of mouse embryoid body. In addition, co-IP analyses showed that STRAP is a putative spliceosome-associated factor. The study reported a relatively small number of AS events that may be regulated by STRAP. The authors also extended the study to *Xenopus* and showed that loss of Strap led to delayed neural tube closure. Overall, this is an interesting study that revealed a novel splicing regulator. However, there are a number of major concerns that need to be addressed.

A major concern is the relatively small number of AS events detected by RNA-seq in response to the loss of Strap. Furthermore, an even smaller number of these AS events had eCLIP peaks in their immediate neighborhood (in the exon or flanking introns). If STRAP is a putative spliceosome-associated factor, there expects to be a much larger pool of its splicing regulatory targets. The small number of AS events is likely due to the fact that only 2 biological replicates were included for each group in the RNA-seq study. Current practice standard is to include 3-5 replicates at least to better estimate sample-to-sample variation. Another reason that could underlie the small number of significant events is the limited analysis method for differential splicing discovery in RNA-seq. It is well known that existing methodologies are limited and different methods could yield very different results. It is strongly recommended that the authors use alternative methods for this analysis, in addition to rMATS, such as MAJIQ, LeafCutter, etc. Outputs of different methods should be compared, and combined in some way to enhance the quality of their results.

To make the point that N2B27-induced EBs molecularly resemble the mouse embryonic germ layer specification, the authors listed some example genes that are differentially expressed in EBs, which are enriched in expected functional categories. Instead of only listing example genes (supplementary fig 3d), the authors should carry out a global comparison between differential gene expression profiles of EBs and those of mouse early germ layer development. To support their conclusion, there needs to be global concordance based on such comparisons.

In EBs, the loss of STRAP induced changes in gene expression. Among genes listed in Supplementary Table 4, why weren't the genes mentioned in the first section (such as *Fgf8*, *Gsc*, *Otx2* and *Shh*) among those that were significantly differentially expressed?

The eCLIP analysis generated a large number of binding peaks of Strap. However, only 4.4% were

associated with annotated AS genes. This result indicates that Strap's main role may not lie with regulation of alternative splicing. Does it more often regulate constitutive exons? That is, is it possible Strap is a necessary factor for maintaining constitutive splicing? In Fig. 4b, for peaks binding to exons or introns, how many peaks are close to constitutive exons? In Fig. 4c and d, were all exons used in the meta-analysis? Did the differential splicing analysis using RNA-seq data allow detection of splicing changes in exons that were not alternatively spliced in the WT cells? Overall, the eCLIP results do not explain much of the AS results from RNA-seq, the reason of which needs to be further investigated.

In the motif analysis using HOMER, what were used as the background exons? The choice of background can dramatically change the results, which needs to be carefully justified.

The authors attempted to related STRAP function with that of other splicing factors. Different analyses eluded to different factors without a consensus (e.g., PRPF3, SF3, SRSF1, SRSF2, SNRPA, U2AF2). A unifying analysis should be conducted to draw clear conclusions on which factors may be directly related to Strap function.

Minor issues:

The terminology used for alternatively skipped exons (SE) should be updated. Sometimes "CA" was used, in other cases, "ES" was used. Neither is standard in the splicing literature. Instead, "SE" should be used to refer to skipped exons (i.e., cassette exons).

The manuscript is very long. Some contents could be condensed/removed from the main text, especially those not directly focusing on the function of STRAP, for example, the first section "Substantial AS events occur during mouse early organogenesis".

Reviewer #3 (Remarks to the Author):

This work highlights a neurodevelopmental role for the serine threonine kinase receptor-associated protein (STRAP) in mouse embryos, embryonic stem cell (mESC) derived embryoid bodies (EBs) with validation in *Xenopus*. The authors use a range of orthogonal models and experimental models to convincingly demonstrate the role of STRAP in neurodevelopment. Overall, this is interesting work and the conclusions are broadly supported by the experimental data provided. There do remain some unanswered questions (e.g. how STRAP binding to the same positions can exert opposing effects in different circumstances), but these are generally acknowledged with plausible hypotheses in the manuscript. Overall I do believe that this work is of the calibre to be published in *Nature Communications* and the authors have done a good job in validation of core findings, orthogonal methods to confirm particular results and a balanced conclusion. Although I'm not suggesting that these should preclude publication, it is noteworthy that there is no actual mechanism presented; i.e. how is

STRAP actually regulating these splicing events. On balance, with the appropriate revisions, I'm supportive of the work being published. Specifically, it would be further strengthened by the authors considering / addressing the following:

1) The first paragraph in the introduction should state explicitly that it is referring to mouse development

2) The rationale for choosing the timepoints is morphological change. However, it is likely that the transcriptional program driving this precedes physical changes. Therefore, could the story be richer / more comprehensive by having an earlier timepoint as a comparator? Also I believe that these are data generated from another previously published study (Chen et al 2004) that have been re-analysed here. Of course, this does not detract from the current findings but it would be more transparent to state this explicitly i.e. a previous study demonstrated X (Chen et al 2004). We reanalysed the data to confirm X but also extend this observation to show Y.

3) 'To delineate the alteration in molecular features during this transitional course, we applied global transcript profiling of the above two stages.' – please describe what the sample was (i.e. homogenized tissue, specific micro-dissected tissue, purified cell types?)

4) 'Strap – /– mutants failed to express several early brain developmental markers (such as Fgf8, Gsc, Otx2 and Shh18) as early as E8.5, (Fig.2c); however, other brain markers (i.e. En1, Hoxb1 and Six3) were not affected by the ablation (Supplementary Fig. 2b)' – Is this because specific brain regions are not formed or is it because the cells within the region do not express the correct markers? Again, it was previously reported that no differences exist at early embryonic stages, but by 9.5 (after organogenesis) there are striking differences. Therefore organogenesis may begin unaffected and the impact of STRAP comes into play later in development. Embryonic lethality occurs between E10.5 and E11.5. This has been previously published and Chen 2004 and should be cited clearly at this point. It will be important to determine at which point the markers displayed in Fig2C,D are lost, as they study the expression at the time point in which embryonic lethality is occurring.

5) Association of STRAP with spliceosomal factors does not necessarily imply that it is involved in the process of spliceosome assembly. STRAP could be sequestering splicing factors away for example. How can these correlative data (co-IP, IF) be taken further to prove that STRAP itself is actually crucial for spliceosome assembly as the authors propose?

6) 'As expected, little is altered in transcriptome profiling of Strap-KO ESCs relative to that of their counterpart WT cells (data not shown)' – Can these data please be shown together with transcript level differences between STRAP-KO and WT. Similarly, in existing tissue from these experiments, it would strengthen the study to see the results validated in the early embryo to see if they are consistent between the EB model and in vivo.

7) Have the EBs been characterized at all? Germ layer markers, percentage of neural cells, regional

identity, capacity for terminal differentiation and acquisition of electrophysiological properties would be helpful to include

8) ‘...indicating these variants have potential biological functions instead of undergoing nonsense-mediated mRNA decay’ – This could be tested formally by genetically and/or pharmacologically inhibiting NMD and looking for a change in the transcript level

9) Stage-specific description of alternative splicing programs has recently been investigated using human stem cell approaches and found that intron retention was a predominant event early in lineage restriction, whilst exon skipping dominated at later stages. It would be useful for readers to contextualise the present study with this work perhaps in the discussion.

10) ‘In contrast, there was a low number of AS outcomes (n=130) affected by STRAP in ESCs (data not shown), raising the possibility that STRAP might mediate splicing in a cell type-specific manner.’ – Please show the data.

11) The authors can begin to address the cell type specificity of STRAP-mediated splicing effects by comparing their data from different stages of lineage restriction (e.g. iPSC, EB, terminal differentiation)

12) Supp Figure 2 A FISH images used to show the expression of STRAP RNA in the embryo could be clearer (particularly the dorsal image).

13) Supp Figure 2 C The teratoma validation is morphologically consistent but markers for the 3 germ layers should be examined by IHC.

14) Supp Figure 2 E: The IgG is not as clean as expected and why are so few peptides detected?

15) The claim that STRAP doesn’t interact with DDX15 in the nucleus could be substantiated by absence of DDX15 on STRAP Co-IP and vice versa. The colocalisation by ICC is only partially convincing

16) The team could have use a conditional knock out in the nervous system to overcome embryonic lethality. However, I do not expect the authors to do this as it is a major undertaking and I feel not a reasonable request for this paper given i) the conceptual advance that has been presented and ii) time frame of a rebuttal.

17) Some in vitro binding assay validation of the motifs would be useful, either generating a mutant and showing that STRAP binding is lost, or using a blocking agent (e.g. MIXmer ASO) to mask it.

18) Suggesting STRAP interacts with other splicing regulators seems unsubstantiated.

19) Fat1 is one of the least affected genes following STRAP loss, why did the authors choose this one and not another?

20) Can the authors perform WB to evaluate an increase in specific protein isoforms resulting from their finding of exon inclusion for example.

21) Why is the length of the exon displayed as a log₁₀ value? The findings in the majority of figure 6 appear purely descriptive and could be moved to supplementary. The motif analysis could have been supplementary when showing the interaction of STRAP to DNA.

22) There is limited information on whether STRAP is in a complex with these splicing regulators or is simply nearby?

Minor changes:

1) Bottom of page 3: Gene oncology (GO) should be Gene ontology (GO)

2) What were the number of biological and technical repeats?

3) 'We further compared splice site scores in responsive and unresponsive CAs. We observed that both STRAP-enhanced and -repressed exons has weaker 3' splice site (ss) than those found in unresponsive CAs (unpaired Wilcoxon test, enhanced $P < 0.0001$, repressed $P < 0.067$), and that enhanced CAs had even weaker 3'ss than repressed ones (Fig. 3d,e).' – Have instead of has

4) Figure 3D can be moved to supplementary, it does not seem important enough to the overall narrative to justify being in the main text.

5) The schema (last panel of last figure) would benefit from a key within the actual figure.

6) 'the total transcripts of these genes did not change much' seems quite casual and should be rewritten.

Reviewer #4 (Remarks to the Author):

In this manuscript, Jin et al. identified STRAP as a novel regulator of alternative splicing (AS) in early mouse development. In developing mouse embryos, the authors uncovered the previously uncharacterized AS signatures in between E8 and E9 mouse embryos when embryos undergo organogenesis including massive neurogenesis and brain patterning. Using Strap knockout embryoid body (EB) differentiation model, the authors found stage specific AS patterns regulated by direct Strap binding and identified the Strap binding motifs. Moreover, using amphibian *Xenopus* model, authors

demonstrated that the AS regulation by Strap is evolutionarily conserved and important for neural development.

Major Points

1. The AS of early developmental genes show highly stage specific patterns. Thus the developmental stages should be carefully considered (and need to be well-presented in the figure panels to be easily recognized). In EB differentiation model, the authors found 454 AS events in the Strap knockout EBs (Fig 3) at D9 (9 days of differentiation). However, at gene level analysis (Fig 5), authors compared splicing patterns of *Nnat* and *Mark3* along the differentiation until D14 and found that Strap affects their AS events somewhere in between D11 and D14. Are there any examples of genes whose ASs are mainly regulated at around D9 EBs? Testing and validating individual AS patterns at this stage (D9) would strengthen the authors' hypothesis that the altered AS pattern in the Strap KO embryos might have caused lethality beyond E10.5 (and impaired neurogenesis in D11-D14 EB model shown in Supplementary Fig 5j).

More specifically, according to the data in Supplementary Fig 5j, KO EBs at D14 might have been already lost its potential to further differentiate (exemplified with the impaired gene expressions of *Ncam1*, *Ntf3* and *Map2*). Therefore, it is reasonable to consider that D14 KO EBs are not in the same differentiation states with WT D14 EBs but rather arrested earlier along the differentiation course. Therefore, the altered splicing patterns of D14 KO EBs do not fully support the idea that Strap directly regulates AS events at later stages (i.e., D14). Authors should validate that the altered AS patterns in Strap KO are not due to the indirect effects of developmental arrest.

2. Again, I think the direct correlation between Strap binding and the AS regulation is largely missing. Since Strap KO might lead to developmental arrest in EB differentiation models (see Supplementary Fig 5j), I would suggest the authors to do the transient knockdown experiments using inducible shRNA expression at D14 EB (or inducible KO model if possible) and see whether Strap knockdown at this specific stage (or at earlier stages) could still affects AS of some of the neuronal target genes (such as *Nnat* and *Mark3*).

3. The *Xenopus* Strap morphant phenotype of delayed neural tube closure is too vestigial and insufficient to justify the authors' claim of evolutionary conservation of the mechanisms and functions of Strap. I think the Strap morphant *Xenopus* embryos should be re-analyzed for expressions of early neural genes (pan-neural, neuronal or region specific neural marker genes) or for Strap target candidate genes identified in mouse EB model. The authors previous paper (also cited as #17) already showed that Strap knockdown causes impaired neural/body patterning. Perhaps neutralized animal cap (e.g., by injecting *Noggin*) model can be utilized for gene expression analysis and AS changes instead of the whole embryo samples. This might particularly helpful since the splicing defects shown in Fig 6g are not very clear enough whether the changes are significant. Can some statistical tests be applied to RTPCR data? Are there any (more) Strap target candidates that can (theoretically) explain the Strap morphant phenotypes?

Minor points

1. In Fig 1c, was there any statistical measures to determine that ES and MXE are mainly regulated? What aspects indicate that ES and MXE are substantially increased along the developmental progression?

2. The authors argue that “Loss of STRAP selectively affected expression of brain-regional markers” (p5, line 176-177). However, it is unclear how the reduction of Strap sensitive early brain marker genes (such as *Fgf8*, *Gsc*, *Otx2* and *Shh*) did not affect other brain markers (*En1*, *Hoxb1* and *six3*) at E8.5 while it led to the failure of further developmental progression and later germ layer marker gene expressions (such as *Irx3*, *Nkx2.5*, *Gata6* and *Tcf15*) in E9.5 KO embryos. It seems that Strap deficiency leads to overall developmental arrest by affecting widespread genes throughout all three germ layers rather than specifically affecting a subset of brain specific target genes. Authors should discuss about the results in more detail.

3. Immunocytochemistry analysis of Strap localization indicated its co-localization with several splicing factors but not with spliceosome disassemble protein DDX15 (Fig 2i and Supplementary Fig 2f). However, the immunostaining signals of STRAP and other proteins seem often over-saturated especially in the case which the authors argue the co-localization. Can the authors provide under saturated images to justify co-localizations better?

We would like to thank the reviewers for the helpful comments on our manuscript. We have addressed all the points raised and feel that our manuscript is now much improved with the changes in the text and additional new experiments. We have discussed each point in detail as follows:

Response to Comments of Reviewer 1

General Comment: “This manuscript by Jin et al. shows STRAP knockout affects many alternative splicing events and early organogenesis during embryonic development. The authors first found Strap KO exhibits dysmorphogenesis as early as E8.5-9 and lethality at E10.5. Transcriptome analysis showed a major defect in alternative splicing regulation but modest transcriptional abnormalities. Proteomics on Strap co-IP samples identified spliceosome components as Strap’s major interacting partners in the nucleus, suggesting STRAP be a splicing regulator. The authors used eCLIP to locate STRAP’s RNA binding sites and found many in introns. Although the informatics data are largely descriptive, many are derived from in vivo samples, which is a significant strength. The manuscript is easy to read. As presented, the novelty of the paper is proposing STRAP as an alternative splicing regulator. Some clarifications are needed (see below). The novelty can be enhanced with mechanistic studies of how STRAP regulates splicing and/or functional analysis of STRAP’s targets.”

Response: *We appreciate the encouraging comments of the reviewer regarding novelty and strength of this interesting study. The concerns raised by this reviewer are being addressed as follows.*

Major Comments

1. Comment: “Fig 2: is the STRAP-spliceosome interaction RNA dependent? Because TFIIC complex is pulled down, the interaction could be even DNA dependent? The authors can answer these questions with DNase and RNase treatment of cell lysates. If the authors decide to pursue the mechanistic aspect, mapping of interaction domains and functional consequence of interaction mutations should be tested.”

Response: *We agree with the reviewer that it is important to explore whether the interactions between STRAP and spliceosome components are in an RNA-dependent manner, which is being addressed here. Overall, the results show that the association of STRAP with U2 proteins remained unchanged or enhanced (new Fig. 3c) after treatment with RNase. In contrast, other U proteins bind to STRAP in the presence of RNA (new Fig. 3c). We then have focused on whether and how STRAP participates in steps of U2 snRNP biogenesis and found that STRAP is involved in the assembly of 17s U2 snRNP complex (new Fig. 7). Additionally, we indeed have observed the contact between STRAP-TFIIC110 and DNA (new Supplementary Fig. 3e), suggesting that STRAP might play a role in the co-transcriptional process. We will explore this possibility in the future work. Regarding the binding domain specificity of STRAP and functional activity in terms of splicing events, we have performed the gain-of-function assay using a well-characterized SR140-dependent UPF3A minigene reporter. We observed that STRAP, specifically through its third WD40 repeat domain, plays a critical role in UPF3A exon4 splicing (new Fig. 7h). We also have obtained similar conclusion using other two minigenes (new Supplementary Fig. 7c,d). Collectively, these results support the notion that STRAP is required for the recruitment of certain splicing factors to spliceosome complex.*

2. Comment: “Does STRAP directly interact with RNA? CLIP is powerful but still has false positives and limitations. The author can use EMSA to measure STRAP binding to RNA with the CLIP motifs or any proposed STRAP gene targets (fig 4e).”

Response: *We agree with the reviewer and this is important since radiolabeled STRAP binds to RNA complexes in mouse embryos and EB cells in an RNA-dependent manner (new Fig. 5a). We added new data from RNA EMSA assays with controls showing that STRAP directly binds with the Nnat transcript at its 3'UTR (~23 to 52 bp downstream of the stop codon) via two predicted binding motifs (new Fig. 5f). Therefore, combining the in vivo and vitro results, we demonstrate that STRAP physically interacts with RNA.*

3. Comment: “RNA-seq targets and eCLIP-Seq targets do not substantially overlap (fig 5a). This does not necessarily invalidate the authors' conclusions, but the authors should provide a thorough explanation.”

Response: *This is a good point. The overlap percentages we presented in the original manuscript were not ideal (previous Fig. 5a). As suggested by the reviewer #2, we have now re-run RNA-seq data using MAJIQ tool for analyzing both classic and non-classic alternative splicing events. Totally, we have obtained 1,462 altered local splicing variants (LSVs) (out of 1163 genes, new Fig. 4h) upon deletion of STRAP in 9-day-old EBs (new Supplementary Table 7). Together with the data from rMATS tool, the overlap percentages between STRAP splicing targets and its eCLIP targets were greatly increased (from 4.4% to 15.6%). More specifically, we found that 15.6% (319 out of 2047) of STRAP binding events are associated with its regulated AS genes (both classic- and non-classic types) and 22.4% (319 out of 1423) of unique genes with AS events in 9-day-old EBs are linked to STRAP binding (new Fig. 6a). This supports our hypothesis that STRAP developmentally regulates certain AS events through direct binding. Updated results have been included in the revised manuscript (new Fig. 6a).*

4. Comment: “To show STRAP as a splicing regulator, the author can test whether removal or mutation of the STRAP binding motif would abrogate STRAP's regulation of a specific exon. The authors can pick a few to test from the 90 candidates (fig 5a). These can be done in minigenes or cell lines.”

Response: *We have conducted these experiments and hope that the addition of this new data strengthens our conclusions. We have now constructed two minigenes containing Nnat (exon1-3) and Ppp2r2d1 (exon3-5), and conducted rescue assays in MEF Strap WT and KO cells. The reason we chose these two transcript sequences is that their specific exon skipping events correlated with the presence of STRAP. RT-PCR assays show that STRAP promoted the exclusion of Nnat cassette exon 2 and the inclusion of Ppp2r2d cassette exon 4 (new Supplementary Fig. 7c and d). Using the domain-truncated vectors, we also demonstrated that the critical sequence of STRAP required for activating splicing locates on its third WD-40 domain (new Supplementary Fig. 7c and d).*

5. Comment: “The phenotypic characterization of the KO mice is rather limited and need to be elaborated”.

Response: *We have shown that deletion of STRAP leads to mouse embryo early lethality at E9.5 to E10.5 dpc. We further compared the morphologic phenotype at E7.5 and E8.5. As shown in new Fig. 2a, there are no obvious differences of embryonic morphology between Strap*

WT and KO mouse embryos. Here, we added more annotation for E7.5 embryo (new Fig. 2a) and revised the description as follows (page 4):

'However, E9.5 *Strap*^{-/-} embryos are small in size with delayed development as compared to WT littermates, namely no body turning with truncated frontonasal region, dilated heart cavity (Fig. 2a), and open neural tube (data not shown), consistent with a previous report of gene trap mutagenesis study¹⁹.

6. Comment: "Although ascribing a phenotype to specific splicing changes are challenging in this context, the authors need to show some STRAP-controlled splicing changes are functionally relevant to implicate the misregulated splicing underlies the phenotype. For example, do two isoforms have different biochemical functions? Does the splicing change impact cell physiology?"

Response: *We agree with the reviewer that some experiments should be performed to verify the biological function of the STRAP targeted isoforms. But as the reviewer mentioned, we feel that it is hard to perform functional assays in cell-based system. In that case we have to use morpholino oligos (MOs) to block the intron sequences upstream of STRAP targeted exons. As MOs have a relatively low efficiency to be delivered into the primary cells, we may see little difference in transcription levels after MOs treatment. Despite the in vitro experimental limitations, we have successfully demonstrated that STRAP-regulated Clk1 isoforms can be functionally distinct in Xenopus model. As shown in new Fig. 8fg, complete blockage of both Clk1 isoforms impair early Xenopus mesoderm development, and selective alteration of the ratio of the two Clk1 products (more exon 4 skipping) led to neural development defects.*

7. Comment: "An important question is whether STRAP is a general splicing regulator or a lineage/stage specific regulator. STRAP's expression patterns and its KO/KD in a context other than germline deletion can answer this question. The latter also helps to mitigate the concern of indirect effects from germline deletion for 8 days."

Response: *We agree with the reviewer and have now provided new AS profiling from tissues of conditional (intestinal-specific) Strap knockout mice (new Supplementary Table 6). Overall, together with the data from ESCs (new Supplementary Table 6), there is a significantly low number of AS events affected by deletion of Strap either in mouse ESCs (ESC vs. EB, $p < 0.0001$) or in mouse intestinal tissues (cKO vs. EB, $p < 0.0001$). These results suggest that STRAP is a splicing regulator in a lineage-specific manner rather than a general one. Please see page 7 for the revised writing.*

Minor Points

8. Comment: "Fig. 1a: please use fold changes or expression values in heatmap. Z-scores are influenced by variances and not really informative."

Response: *Considering gene isoforms have their own transcript expression, we have removed previous Fig. 1a and replace it by a new table to show global fold changes at the isoform level during mouse early embryo organogenesis (new Supplementary Table 1).*

9. Comment: "Fig. 1f: the scamp5 gel image is not consistent with the browser tracks."

Response: *We thank the reviewer for this point. It is now fixed (new Fig. 1e).*

10. Comment: “Fig2g: what does each sub-network represent? What does a link between proteins mean? PPI?”

Response: We apologize for the lack of clarity in our original description. We used COMPLETEAT tool to analyze our proteomic data. This tool enables to predict protein complex based on the protein-protein interaction (PPI) networks. We have revised the text as follows (page 5):

‘Additionally, the assigned ranked scores by COMPLETEAT tool²⁹ showed 30% of hits (45 out of 152) participate in multiple predicted networks between splicing protein interactions (Fig.3b and Supplementary Fig.3c)’.

11. Comment: “Fig3h-i: Variability between the two biological replicates appears very high. how robust are these targets controlled by STRAP? For example, plekhg4 in fig3i does not appear to be regulated in Strap KO.”

Response: We apologize for the mistake to present unchanged comparison for genes (such as plekhg4, Atxn1 and Evc), this has now been rectified (new Supplementary Fig. 4g-j). Even through we indeed show several AS exons (such as Myo9a, Camk2d-1 and Tcf7l2-1) had biological variation in duplicate samples, they exhibit same altered trends, which were either inclusion or exclusion compared to respective control. Overall, the heatmaps (new supplementary Fig.4 g-j) show highly repeated close values between each duplicate. We also conducted RT-PCR assay to assess the accuracy for analyzing AS events (new Supplementary Fig. 4k). Together, these support the robustness of our findings.

12. Comment: “Supp fig3d: looks like a large variance between biological replicates?”

Response: It is true that during early mouse embryo development, specially in blastocyte stage, certain genes may exhibit the expression variability between embryos. This is mainly caused by individual asynchronous development. In our studies, we established blastocyst-derived mouse ESC lines (including Strap WT, Het and Homo) from the same pregnant female mouse, which could greatly decrease the individual background in terms of expression profiling, despite they still display somewhat variation between replicates as shown in previous Supplement Fig. 3d. We have now globally analyzed and compared the expression dataset using principle components and hierarchieal clustering analyses and show that EB replicates had high reproducibility (new Supplementary Fig. 4e, f). To avoid confusion of the reviewers and readers, we have removed previous Supplementary Fig. 3d from the revised manuscript.

Response to Comments of Reviewer# 2

General comment: “This manuscript reports a study of STRAP as a novel regulator of alternative splicing in early mouse embryos. The authors carried out global studies of STRAP using RNA-seq, eCLIP-seq, complemented by experimental validations and individual assays, in knockout models of mouse embryoid body. In addition, co-IP analyses showed that STRAP is a putative spliceosome-associated factor. The study reported a relatively small number of AS events that may be regulated by STRAP. The authors also extended the study to Xenopus and showed that loss of Strap led to delayed neural tube closure. Overall, this is an interesting study that revealed a novel splicing regulator. However, there are a number of major concerns that need to be addressed.”

***Response:** We appreciate the reviewer's affirmative comments and insightful suggestions that are helpful in improving the manuscript.*

Major Comments

1. Comment: "A major concern is the relatively small number of AS events detected by RNA-seq in response to the loss of Strap. Furthermore, an even smaller number of these AS events had eCLIP peaks in their immediate neighborhood (in the exon or flanking introns). If STRAP is a putative spliceosome-associated factor, there expects to be a much larger pool of its splicing regulatory targets. The small number of AS events is likely due to the fact that only 2 biological replicates were included for each group in the RNA-seq study. Current practice standard is to include 3-5 replicates at least to better estimate sample-to-sample variation. Another reason that could underlie the small number of significant events is the limited analysis method for differential splicing discovery in RNA-seq. It is well known that existing methodologies are limited and different methods could yield very different results. It is strongly recommended that the authors use alternative methods for this analysis, in addition to rMATS, such as MAJIQ, LeafCutter, etc. Outputs of different methods should be compared, and combined in some way to enhance the quality of their results."

***Response:** We thank the reviewer for providing valuable suggestions. First, we have globally analyzed the RNA-seq data from biological duplicates using PCA and HCl approaches, which reveal that EB replicates in either WT or KO group had high reproducibility (new Supplementary Fig. 4e,f). Second, we agree with the reviewer that rMATS tool can sometimes be underpowered. We, therefore, re-analyzed the RNA-seq data using MAJIQ tool, as suggested by this reviewer. We obtained 1,462 altered local splicing variants (out of 972 genes) upon deletion of STRAP in Day 9 EBs (new Supplementary Table 7), which has a significant concordance with the data from rMATS (new Fig. 4h). By combining these two datasets, we obtained 1687 AS events (new Fig. 4h, including classic- and non-classic splicing patterns) regulated by STRAP. This number is greatly increased from 4.4% to 15.6%. In detail, we found that 15.6% (319 out of 2047) of STRAP binding events is associated with its regulated AS genes (both classic- and non-classic types) and 22.4% (319 out of 1423) of unique genes with AS events in 9-day-old EBs are linked to STRAP binding, supporting our hypothesis that STRAP developmentally regulates certain AS events through direct binding. We hope the additional data make our results more informative.*

2. Comment: "To make the point that N2B27-induced EBs molecularly resemble the mouse embryonic germ layer specification, the authors listed some example genes that are differentially expressed in EBs, which are enriched in expected functional categories. Instead of only listing example genes (supplementary fig 3d), the authors should carry out a global comparison between differential gene expression profiles of EBs and those of mouse early germ layer development. To support their conclusion, there needs to be global concordance based on such comparisons."

***Response:** To address the reviewer's concern, we have added a systematic global comparison of our findings with three datasets previously published (new Supplementary Fig. 4e,f). In these figures, the previous findings from induced neuroectoderm-like EBs by Li et al. (Cell Reports, 2020) are shown to be very close to the context of our model; however, the data from either early mouse brain tissues (Ayoub et al, PNAS, 2011) or three germ layer-derived multiple organs in mouse embryo early development (Werb et al, 2014) is far away with our data in*

cluster analysis. We further characterized the EBs during the course of time using molecular markers (New Supplementary Fig. 4c) and electrophysiological assay (new Fig. 6d,e), which show that N2B27-induced EBs molecularly resemble the neuroectodermal lineage commitment and have a terminal differentiation toward matured neuronal cells after long-term induction of ascorbic acid.

3. Comment: “In EBs, the loss of STRAP induced changes in gene expression. Among genes listed in Supplementary Table 4, why weren’t the genes mentioned in the first section (such as Fgf8, Gsc, Otx2 and Shh) among those that were significantly differentially expressed?”

Response: We agree that there is no mention of the above-mentioned genes in previous Supplementary Table 4, in which differentially expressed genes with a P value < 0.01 as well as a \log_2 fold change > 1 were listed. According to the thresholds, there are no significant changes of these genes when compared KO with WT groups at day 9. We have now quantified these markers by qPCR assays at two time points (day 9 and day 11) and found several genes down-regulated by loss of STRAP at day 11, consistent with the *in vivo* data (new Fig. 2c). These results are now included in new Supplementary Fig. 4d.

4. Comment: “The eCLIP analysis generated a large number of binding peaks of Strap. However, only 4.4% were associated with annotated AS genes. This result indicates that Strap’s main role may not lie with regulation of alternative splicing. Does it more often regulate constitutive exons? That is, is it possible Strap is a necessary factor for maintaining constitutive splicing? In Fig. 4b, for peaks binding to exons or introns, how many peaks are close to constitutive exons? In Fig. 4c and d, were all exons used in the meta-analysis? Did the differential splicing analysis using RNA-seq data allow detection of splicing changes in exons that were not alternatively spliced in the WT cells? Overall, the eCLIP results do not explain much of the AS results from RNA-seq, the reason of which needs to be further investigated.”

Response: We appreciate reviewer’s suggestion of using alternative method to analyze splicing events. In previous Fig. 5a, the overlap percentages between STRAP peaks and annotated AS genes were not ideal. We have now reanalyzed the intersected percentages using combined AS genes (from rMATS and MAJIQ) with eCLIP genes, which results in the increased percentages from 4.4% to 15.6% and has been updated in new Fig. 6a.

We show STRAP plays potential roles in both constitutive and alternative exons. Because (1) we have leveraged STRAP RNA-binding reads along all exon/intron and intron/exon borders. In comparison to the input, STRAP signals in samples were dramatically enriched within ~50 nucleotides upstream of the 5’ss and ~25 nucleotides downstream of the 3’ss (previous Fig. 4c). Thus, the binding of STRAP preferentially in close proximity to both 5’ and 3’ of splicing sites suggests its potential role in demarcating exonic regions; (2) 22.4% of AS genes (319 out of 1422) could be directly regulated by STRAP between eCLIP peaks and AS hits (new Fig. 6f); and (3) On average, we obtained 3076 binding peaks close to exons (within 300 nt region upstream or downstream of exon), among which 964 peaks were fallen on annotated alternative exons and 2112 associated with constitutive exons. We have added this information in the new Supplementary Table 9 and accordingly revised the text (page 9). Taken together, these data strongly support our hypothesis that STRAP developmentally regulates numerous AS events through direct binding.

In previous Fig. 4c and d, we performed meta-analysis based on all exons obtained by eCLIP-seq. Assuming the value of PSI is 1 in terms of the exon without alternative splicing, we computationally extracted 39 cases (17.5%, out of 223) from rMATS data, in which exon

skipping only occurred in STRAP KO EBs. In other 194 cases, we see that WT cells have either skipping exon or inclusion one relative to KO parallels.

5. Comment: “In the motif analysis using HOMER, what were used as the background exons? The choice of background can dramatically change the results, which needs to be carefully justified.”

Response: We thank the reviewer’s point. The background sequences were automatically selected by HOMER from the genome sequences based on GC% content (Heinz, et al, Mol Cell, 2010), as we did not define the background sequences. The parameters for the main script “findMotifsGenome.pl” in HOMER software were: findMotifsGenome.pl <peak/BED file> <genome> <output directory> -size given –rna –mask. Additionally, RNA EMSA assays provide more evidence that STRAP directly binds the Nnat transcript at its 3’UTR via two HOMER predicted binding motifs (new Fig. 5f).

6. Comment: “The authors attempted to relate STRAP function with that of other splicing factors. Different analyses eluded to different factors without a consensus (e.g., PRPF3, SF3, SRSF1, SRSF2, SNRPA, U2AF2). A unifying analysis should be conducted to draw clear conclusions on which factors may be directly related to Strap function.”

Response: We agree that more detail about the association of STRAP with spliceosome complex should be added. The RNase treatment did not interfere with STRAP binding to U2 snRNP components (new Fig. 3c) indicating that these interactions depend on protein-protein contact. We therefore focus on the relationships between U2-specific proteins and STRAP. CO-IP assays reveal that STRAP regulates the assembly of 17S complex at the stage of U2 snRNP biogenesis (New fig.7c-f). Please also see our response to reviewer# 1 comment 1 for additional details on this point.

Minor Comments

7. Comment: “The terminology used for alternatively skipped exons (SE) should be updated. Sometimes “CA” was used, in other cases, “ES” was used. Neither is standard in the splicing literature. Instead, “SE” should be used to refer to skipped exons (i.e., cassette exons).”

Response: We agree with the reviewer and have maintained the consistency of using “SE” in the revised manuscript.

8. Comment: “The manuscript is very long. Some contents could be condensed/removed from the main text, especially those not directly focusing on the function of STRAP, for example, the first section “Substantial AS events occur during mouse early organogenesis.”

Response: We have now condensed and reorganized result sections in the revised manuscript.

Response to Comments of Reviewer# 3

General Comment: “This work highlights a neurodevelopmental role for the serine threonine kinase receptor-associated protein (STRAP) in mouse embryos, embryonic stem cell (mESC)

derived embryoid bodies (EBs) with validation in *Xenopus*. The authors use a range of orthogonal models and experimental models to convincingly demonstrate the role of STRAP in neurodevelopment. Overall, this is interesting work and the conclusions are broadly supported by the experimental data provided. There do remain some unanswered questions (e.g. how STRAP binding to the same positions can exert opposing effects in different circumstances), but these are generally acknowledged with plausible hypotheses in the manuscript. Overall, I do believe that this work is of the caliber to be published in *Nature Communications* and the authors have done a good job in validation of core findings, orthogonal methods to confirm particular results and a balanced conclusion. Although I'm not suggesting that these should preclude publication, it is noteworthy that there is no actual mechanism presented; i.e. how is STRAP actually regulating these splicing events. On balance, with the appropriate revisions, I'm supportive of the work being published". Specifically, it would be further strengthened by the authors considering / addressing the following":

Response: We thank the reviewer for positive comments and for providing insightful suggestions that provided opportunities to improve the manuscript.

Major Comments

1. Comment: "The first paragraph in the introduction should state explicitly that it is referring to mouse development".

Response: We have revised the "Introduction" part and referred to mouse development (page 1).

2. Comment: "The rationale for choosing the time points is morphological change. However, it is likely that the transcriptional program driving this precedes physical changes. Therefore, could the story be richer / more comprehensive by having an earlier timepoint as a comparator? Also I believe that these are data generated from another previously published study (Chen et al 2004) that have been re-analysed here. Of course, this does not detract from the current findings but it would be more transparent to state this explicitly i.e. a previous study demonstrated X (Chen et al 2004). We reanalysed the data to confirm X but also extend this observation to show Y."

Response: We agree with the reviewer that, as originally written, it is not clear why we chose 9-day-old EBs to perform a serial of experiments. Our rationale is as follows: (1) this timepoint relatively matches the mouse embryonic day on which the morphology of STRAP KO embryo is severely impaired. (2) Based on our culture protocol, EBs could undergo the stage of neuroectoderm lineage on day 9. As the reviewer's request, we quantified the expression of markers at different timepoints. The results show that the expression of neuroectoderm-related genes was comparable between two groups at day 6-9 and brain developmental markers appeared as early as day 9. We have now included these results in new Supplementary Fig. 4d and revised the text accordingly. We also have rearranged the text for the citation Chen's paper (page 4). Please note that in this paper authors only show the Strap KO mouse embryo lethality after E10.5 and describe the alteration in morphology upon Strap deletion. There is no more information shown in this paper to explore the mechanism at cellular and molecular levels.

3. Comment: " 'To delineate the alteration in molecular features during this transitional course, we applied global transcript profiling of the above two stages.' – please describe what the sample was (i.e. homogenized tissue, specific micro-dissected tissue, purified cell types?)."

Response: We have revised the text as follows (page 3):

'To delineate the molecular features of mouse embryos during post-gastrulation (E8.0) to early organogenesis (E9.0), we applied global transcript profiling using whole mouse embryos at these two stages'.

4. Comment: “ ‘Strap – /– mutants failed to express several early brain developmental markers (such as Fgf8, Gsc, Otx2 and Shh18) as early as E8.5, (Fig. 2c); however, other brain markers (i.e. En1, Hoxb1 and Six3) were not affected by the ablation (Supplementary Fig. 2b)’ – Is this because specific brain regions are not formed or is it because the cells within the region do not express the correct markers? Again, it was previously reported that no differences exist at early embryonic stages, but by 9.5 (after organogenesis) there are striking differences. Therefore organogenesis may begin unaffected and the impact of STRAP comes into play later in development. Embryonic lethality occurs between E10.5 and E11.5. This has been previously published and Chen 2004 and should be cited clearly at this point. It will be important to determine at which point the markers displayed in Fig. 2C,D are lost, as they study the expression at the time point in which embryonic lethality is occurring.”

Response: The reviewer raises an interesting point. Morphologic evaluation of Strap-/- embryos (E8.5) shows that they are normal in their own body proportions and orientation. So we think the down-regulated expression of the early brain developmental markers (including Fgf8, Gsc, Otx2 and Shh18) is mainly caused by loss of STRAP in region-specific cells. However, we cannot exclude the possibility that low levels of these genes in Strap-/- mutants were present due to the delayed development of curtailed brain regions. Combined with our previous study in Xenopus (ref 18) and current mouse embryo model, we conclude that STRAP contributes its role in this regard between the late gastrulation and early organogenesis stages. We have quantified and compared the above markers using E7.5 embryos (new Supplementary Fig. 2b). Our data show that germ layers markers were comparable between WT, Het and KO. We have rewritten the results section as follows (page 5).

'Strap deficiency initially has a negative impact on mouse embryo fore- and midbrain development at E8.5 and subsequently causes a uniform and constant developmental delay from E9.5.'

We have changed the write up in results section for citing observations from Chen, 2004, Nature genetics (page 4).

5. Comment: “Association of STRAP with spliceosomal factors does not necessarily imply that it is involved in the process of spliceosome assembly. STRAP could be sequestering splicing factors away for example. How can these correlative data (co-IP, IF) be taken further to prove that STRAP itself is actually crucial for spliceosome assembly as the authors propose?”

Response: We agree that more detailed mechanisms about how STRAP mediates the biogenesis of spliceosome complex should be further explored. We have now included new co-IP data to provide evidence that STRAP involves in the assembly of 17S U2 snRNP complex. These results are now included in new Fig. 7c-f. Please also see our response to reviewer# 1 comment 1 for additional details on this point.

6. Comment: “ ‘As expected, little is altered in transcriptome profiling of Strap-KO ESCs relative to that of their counterpart WT cells (data not shown)’ – Can these data please be shown together with transcript level differences between STRAP-KO and WT. Similarly, in

existing tissue from these experiments, it would strengthen the study to see the results validated in the early embryo to see if they are consistent between the EB model and in vivo.”

Response: We have now added new Supplementary Table 4 to address this comment. We performed qPCR to quantify several markers in EBs and obtained the consistent results similar to those from in vivo studies. Please see the response to Reviewer #2 comment 3.

7. Comment: “Have the EBs been characterized at all? Germ layer markers, percentage of neural cells, regional identity, capacity for terminal differentiation and acquisition of electrophysiological properties would be helpful to include.”

Response: We agree with the reviewer and have systematically compared our findings with previously published datasets. We have shown our model systems have similar gene profiling as previously reported data from induced neuroectoderm-like EBs (ref 33). We have further characterized the EBs during the course of time using molecular markers and found that EBs mostly mimic the development of mouse front- and mid-brains after long-term treatment with ascorbic acid (new Supplementary Fig. 4d). Additionally, both RNA-seq and qPCR of 9-day-old EBs failed to detect the early hindbrain markers (i.e. *En1* and *Hoxb1*). Flow cytometry revealed that the high percentage of CD24⁺/CD56⁺ neuronal cells in WT EBs at day 14 compared to corresponding KO EBs (new Supplementary Fig. 6g). We have also assessed the terminal differentiation of EBs regarding electrophysiological properties and included the data in new Fig. 6d,e.

8. Comment: “ ‘...indicating these variants have potential biological functions instead of undergoing nonsense-mediated mRNA decay’ – This could be tested formally by genetically and/or pharmacologically inhibiting NMD and looking for a change in the transcript level.”

Response: We appreciate the reviewer’s suggestion. We have treated EBs with NMD inhibitor cycloheximide (Durand, et al, *J Cell Biol*, 2007; Dang, et al, *J Biol Chem*, 2009) and examined the levels of 16 spliced transcripts from 8 genes that were randomly chosen from previous Supplementary Fig. 3j. Out of 8, 3 genes (*Usp40*, *Nploc4* and *Lmo7*) with their short isoforms (skipping exon) did not undergo NMD; however, in rest 5 genes (*Clock*, *Eef1d*, *Pms2*, *Mllt10* and *Usp45*), their short mutants were targeted by NMD pathway. We observed these results in both WT and KO groups. So we are cautious not to conclude that these variants have potential biological functions instead of undergoing nonsense-mediated mRNA decay. As shown in new Supplementary Table 4, the gene expression profiling from RNA-seq usually provides the transcripts information matching known annotations (i.e., transcripts from a published database of known genes), and it is more significant if the known transcripts represent a higher proportion of all the transcripts. Based on this, we found genes with changed exons displayed little differential expression upon STRAP deletion. On the other hand, our study is focusing on how STRAP involves in the substantial transcriptional diversity at exon levels during neuroectoderm differentiation. Taken together, we have deleted the above write-up in our revised text.

9. Comment: “Stage-specific description of alternative splicing programs has recently been investigated using human stem cell approaches and found that intron retention was a predominant event early in lineage restriction, whilst exon skipping dominated at later stages. It would be useful for readers to contextualize the present study with this work perhaps in the discussion.”

Response: We have cited a content-related paper and discussed this important point in the revised text (page15).

10. Comment: “ ‘In contrast, there was a low number of AS outcomes (n=130) affected by STRAP in ESCs (data not shown), raising the possibility that STRAP might mediate splicing in a cell type-specific manner.’ – Please show the data.”

Response: We have added the data in new Supplementary Table 6.

11. Comment: “The authors can begin to address the cell type specificity of STRAP-mediated splicing effects by comparing their data from different stages of lineage restriction (e.g. iPSC, EB, terminal differentiation).”

Response: We have now included additional, new AS profiling from conditional (intestin-specific) STRAP knockout mouse tissues (new Supplementary Table 6). Please see our response to Reviewer# 1 comment 7 for additional details on this point.

12. Comment: “Supp Figure 2 A FISH images used to show the expression of STRAP RNA in the embryo could be clearer (particularly the dorsal image).”

Response: We have tried FISH experiments for the expression of STRAP RNA in E9.5 mouse embryo, but the original images are the clearest ones. To show the specificity of anti-sense probes, we have added the new images for the sense probes as negative control (new Supplementary Fig. 2a).

13. Comment: “Supp Figure 2 C The teratoma validation is morphologically consistent but markers for the 3 germ layers should be examined by IHC.”

Response: We have now provided a histological characterization of the teratoma (new Fig. 2e). Results section has been updated accordingly (page 5).

14. Comment: “Supp Figure 2 E: The IgG is not as clean as expected and why are so few peptides detected”?

Response: We acknowledge that due to longer development, the gel showed little background and some non-specific protein bands with IgG. We definitely obtained a number of protein bands due to specific binding with STRAP, which is clear from the raw data of manually enriched STRAP-specific targets as listed in new Supplementary Table 3.

15. Comment: “The claim that STRAP doesn’t interact with DDX15 in the nucleus could be substantiated by absence of DDX15 on STRAP Co-IP and vice versa. The co-localization by ICC is only partially convincing.”

Response: We agree with the reviewer and have now added the data for Co-IP assay. We did not see the interaction between STRAP and DDX15 in the absence or presence of RNase A (new Supplementary Fig. 3d). The text has been updated accordingly (page 5).

16. Comment: The team could have use a conditional knock out in the nervous system to overcome embryonic lethality. However, I do not expect the authors to do this as it is a major undertaking and I feel not a reasonable request for this paper given i) the conceptual advance that has been presented and ii) time frame of a rebuttal.

Response: We thank the reviewer for his/her interest along this line and the understanding that it is not possible to finish those experiments in the time frame. We will definitely explore this in our future work.

17. Comment: “Some in vitro binding assay validation of the motifs would be useful, either generating a mutant and showing that STRAP binding is lost, or using a blocking agent (e.g. MIXmer ASO) to mask it.”

Response: We thank the reviewer for this comment. We have now performed RNA EMSA assays (new Fig. 5f). Please see our response to reviewer# 1 comment 2 for additional details on this point.

18. Comment: “Suggesting STRAP interacts with other splicing regulators seems unsubstantiated.”

Response: We have made major changes by including data in response to this comment. Overall, the result shows that the association of STRAP with U2 proteins remained unchanged or enhanced with RNase treatment (new Fig. 3c). In contrast, other U proteins bound to STRAP in the presence of RNA (new Fig. 3c). Additionally, STRAP involved in the assembly of 17S U2 snRNP complex (new Fig.7). Please see our response to Reviewer# 1 comment 1 for further details on this point.

19. Comment: “Fat1 is one of the least affected genes following STRAP loss, why did the authors choose this one and not another”?

Response: Based on the STRAP-RNA binding map (new Fig. 6b,c), we have performed UV crosslinking and immunoprecipitation (RIP) with anti-STRAP antibody to arbitrarily examine three eCLIP-hit genes, Fat1, Tcf7l2 and Ctnd1. We have now removed results for Fat1 to new Supplementary Fig. 6c.

20. Comment: “Can the authors perform WB to evaluate an increase in specific protein isoforms resulting from their finding of exon inclusion for example”

Response: We appreciate the reviewer’s suggestion. We have carefully checked the commercially available antibodies against the protein isoforms or the antibodies that distinguish two isoforms identified by our RT-PCR assays (new Supplementary Fig. 4k). Unfortunately, only the antibody for specific NAB2 isoform is available. As shown in results below, less short Nab2 isoform was observed in STRAP KO EBs, which is consistent with the change at transcript level.

21. Comment: “Why is the length of the exon displayed as a log10 value? The findings in the majority of figure 6 appear purely descriptive and could be moved to supplementary. The motif analysis could have been supplementary when showing the interaction of STRAP to DNA.”

Response: Because the length of STRAP target exons and introns has a wide range, we have to narrow down the value via log₁₀ conversion. As the reviewer suggested, plots have been moved to new Supplementary Fig. 8a-e. Please see Source data file for detailed raw data. Considering the limited words in text, we have removed the plots for the motif analysis (previous Fig. 6d-e and Supplementary Fig. 6 c-g) from the revised manuscript.

22. Comment: “There is limited information on whether STRAP is in a complex with these splicing regulators or is simply nearby”?

Response: In the original manuscript, we have shown that STRAP is associated with multiple sub-complexes of the spliceosome, including the components of the major snRNPs (U1, U2, U4/U6, and U5) and splicing factors. However, in the revised version, Co-IP assays revealed that STRAP remained associated with proteins after treatment with RNase (such as SR140, SF3A and 3B subunits), suggesting for protein-protein based interactions. Therefore, we now think that STRAP is directly involved in U2 complex through its WD-40 repeat domain (new Fig. 7). The interactions between STRAP and other U components depend on the presence of RNA (new Fig. 3c).

Minor Comments

23. Comment: “Bottom of page 3: Gene oncology (GO) should be Gene ontology (GO).”

Response: We have corrected this.

24. Comment: “What were the number of biological and technical repeats”?

Response: For mouse early embryos, 5-7 biological repeats were used to perform experiments. Two biological repeats for WT and STRAP KO EBs were used in all related experiments, unless otherwise stated. We also used 12-17 biological repeats in Xenopus assays. All experiments were technically repeated at least three times. We have amended the information throughout all the figure legends.

25. Comment: “ ‘We further compared splice site scores in responsive and unresponsive CAs. We observed that both STRAP-enhanced and -repressed exons has weaker 3’ splice site (ss) than those found in unresponsive CAs (unpaired Wilcoxon test, enhanced P<0.0001, repressed P <0.067), and that enhanced CAs had even weaker 3’ss than repressed ones (Fig. 3d,e).’ – Have instead of has.”

Response: Thanks for this point. It is now fixed.

26. Comment: “Figure 3D can be moved to supplementary, it does not seem important enough to the overall narrative to justify being in the main text.”

Response: Previous Fig. 3d was a schematic diagram to define alternative and constitutive exons as well as 5’/3’ splicing sites. Accordingly, we extracted the corresponding DNA sequences and uploaded them to MaxEntScan tool to leverage the splicing scores (as shown in previous Fig. 3e). Therefore, these two subfigures are interdependent. We have now revised them as new Fig. 3d and 3e.

27. Comment: “The schema (last panel of last figure) would benefit from a key within the actual figure.”

Response: We agree that more detail about the mechanism schemas would be useful. We have now updated this figure in new Fig. 8h.

28. Comment: “ ‘the total transcripts of these genes did not change much’ seems quite casual and should be rewritten.”

Response: We agree with the reviewer and it is now fixed as follows:

‘the total transcripts of these genes had little change’

Response to Comments of Reviewer# 4

General Comment: “In this manuscript, Jin et al. identified STRAP as a novel regulator of alternative splicing (AS) in early mouse development. In developing mouse embryos, the authors uncovered the previously uncharacterized AS signatures in between E8 and E9 mouse embryos when embryos undergo organogenesis including massive neurogenesis and brain patterning. Using Strap knockout embryoid body (EB) differentiation model, the authors found stage specific AS patterns regulated by direct Strap binding and identified the Strap binding motifs. Moreover, using amphibian *Xenopus* model, authors demonstrated that the AS regulation by Strap is evolutionarily conserved and important for neural development.”

Response: *We thank the reviewer for providing useful and constructive comments.*

Major Comments

1. Comment: “The AS of early developmental genes show highly stage specific patterns. Thus the developmental stages should be carefully considered (and need to be well-presented in the figure panels to be easily recognized). In EB differentiation model, the authors found 454 AS events in the Strap knockout EBs (Fig 3) at D9 (9 days of differentiation). However, at gene level analysis (Fig 5), authors compared splicing patterns of *Nnat* and *Mark3* along the differentiation until D14 and found that Strap affects their AS events somewhere in between D11 and D14. Are there any examples of genes whose ASs are mainly regulated at around D9 EBs? Testing and validating individual AS patterns at this stage (D9) would strengthen the authors’ hypothesis that the altered AS pattern in the Strap KO embryos might have caused lethality beyond E10.5 (and impaired neurogenesis in D11-D14 EB model shown in Supplementary Fig 5j). More specifically, according to the data in Supplementary Fig 5j, KO EBs at D14 might have been already lost its potential to further differentiate (exemplified with the impaired gene expressions of *Ncam1*, *Ntf3* and *Map2*). Therefore, it is reasonable to consider that D14 KO EBs are not in the same differentiation states with WT D14 EBs but rather arrested earlier along the differentiation course. Therefore, the altered splicing patterns of D14 KO EBs do not fully support the idea that Strap directly regulates AS events at later stages (i.e., D14). Authors should validate that the altered AS patterns in Strap KO are not due to the indirect effects of developmental arrest.

***Response:** We are thankful for this important comment of the reviewer and apologize for not having included the detailed information previously. Using rMATS tool, 454 classical AS events (in 397 genes) were detected in Strap-KO EBs on day 9 (new Fig. 4a). To validate the accuracy of our analyses of AS, we individually measured 28 predicted SEs using 9-day-old EBs (new Supplementary Fig. 4k), demonstrating a high correlation with splicing efficiency (new Supplementary Fig. 4l). The information is now included in revised text.*

Please see our response to comment 2 for additional details on this point.

2. Comment: “Again, I think the direct correlation between Strap binding and the AS regulation is largely missing. Since Strap KO might lead to developmental arrest in EB differentiation models (see Supplementary Fig 5j), I would suggest the authors to do the transient knockdown experiments using inducible shRNA expression at D14 EB (or inducible KO model if possible) and see whether Strap knockdown at this specific stage (or at earlier stages) could still affects AS of some of the neuronal target genes (such as Nnat and Mark3).”

***Response:** We appreciate the reviewer’s suggestion of using inducible knockdown model to verify STRAP-regulated exons. To eliminate the possibility that SE events were directly caused by the differentiation arrest and to see whether Strap knockdown at this specific stage results in the regulation of AS, we developed a Tet-On inducible short hair RNA (shRNA) system against Strap in EBs. Three-day treatment of Doxycycline resulted in an efficient reduction in STRAP expression in 14-day-old EBs (new Fig. 6o). Importantly, compared with control group, inducible STRAP KD cells produced more included exons for Nnat and Mark3 transcripts (new Fig.6o), which is consistent with observations from KO EBs (new Fig. 6i, m).*

3. Comment: The Xenopus Strap morphant phenotype of delayed neural tube closure is too vestigial and insufficient to justify the authors’ claim of evolutionary conservation of the mechanisms and functions of Strap. I think the Strap morphant Xenopus embryos should be re-analyzed for expressions of early neural genes (pan-neural, neuronal or region specific neural marker genes) or for Strap target candidate genes identified in mouse EB model. The authors previous paper (also cited as #17) already showed that Strap knockdown causes impaired neural/body patterning. Perhaps neutralized animal cap (e.g., by injecting Noggin) model can be utilized for gene expression analysis and AS changes instead of the whole embryo samples. This might particularly helpful since the splicing defects shown in Fig 6g are not very clear enough whether the changes are significant. Can some statistical tests be applied to RTPCR data? Are there any (more) Strap target candidates that can (theoretically) explain the Strap morphant phenotypes?

***Response:** We agree with reviewer that the endogenous function of STRAP in Xenopus should be acknowledged better in the results section. In our previous study, to analyze the defects induced by Strap-MO, we performed in situ hybridization (ISH) of a panel of neural markers. For example, Otx2 is a marker for the forebrain and the midbrain. Engrailed (En) is a midbrain-specific gene. Krox20 is expressed in the hindbrain. HoxD4 and HoxB9 are both spinal cord markers. Pax6 is expressed in the spinal cord, the diencephalon, and the eye fields. After injection with Strap-MO in embryos, the above markers were shifted posteriorly relative to the control side at neurula stage, and the expression of Krox20 and Pax6 was also reduced by Strap knockdown (ref 18). We have added the previous conclusion in the revised result section (page14). Here, we take the reviewer’s constructive suggestion and conducted the study specifically in neuralized ectoderm. RNA encoding the neural inducer Noggin was injected into animal regions of 2-cell stage embryos with or without Strap-MO. Ectodermal explants (animal caps) were dissected at late blastula stages and culture in vitro until tailbud stages before RNA*

was extracted. Several neural differentiation markers (i.e. *NeuroD*, *NcAM* and *Ngn*) were reduced at mRNA levels in *Strap*-MO compared to *Noggin*-neutralized parallel (new Supplementary Fig. 8g). Using RT-PCR assay we observed that exon8 skipping of *Cep57* and exon4 exclusion of *Clk1* are STRAP-dependent (new Fig.8d), supporting that STRAP evolutionally controls alternative splicing in *Xenopus* neural tissues. We have now quantitated the data and added below the panels, as advised by the reviewer. To explore functional consequence of alternative splicing of STRAP targets, we focused on *Clk1*. Injection into dorsal animal region with MO against *Clk1* splicing acceptor site at exon 4 resulted in embryos displaying neural tube closure defects (new Fig.8g). Importantly, this phenotype was different from that when *Clk1* ATG-MO was used. The latter induced severe gastrulation defects with the morphant embryos displaying exposed mesendoderm at the tailbud and tadpole stages (new Fig. 8g). Taken together, the results imply that while complete blockage of both *Clk1* products impaired early *Xenopus* mesendoderm development, selective alteration of the ratio of the two *Clk1* products led to neural development defects. To see whether *Clk1* is sufficient to mediate the effects of *Strap*-MO on early *Xenopus* embryogenesis, we attempted the rescue experiments with either *Clk1* splicing MO or *Clk1* RNA. Both ectopic expression of *Clk1* (the long form) and the alteration of *Clk1* splicing products results in aggravated developmental defects when compared with *Strap*-MO alone. This suggests that neural development may be very sensitive to the ratio of the two *Clk1* products so that either increase or decrease the ratio can lead to compromised development of neural derivatives (e.g. head structure). Alternatively, it is also possible that other *Strap* targets cooperate with *Clk1* to control neural development in *Xenopus*.

Minor Comments

4. Comment: “In Fig 1c, was there any statistical measures to determine that ES and MXE are mainly regulated? What aspects indicate that ES and MXE are substantially increased along the developmental progression?”

Response: We apologize for writing the confusing sentence to present the results. There is no comparison of AS events along the developmental progression in this study. We have included the results for mouse embryo at E9.0. We have now changed the text as follows (page 3).

‘Interestingly, the regulated AS events mainly distributed in MXE and SE categories, (42% and 38%, respectively).’

5. Comment: “The authors argue that “Loss of STRAP selectively affected expression of brain-regional markers” (p5, line 176-177). However, it is unclear how the reduction of *Strap* sensitive early brain marker genes (such as *Fgf8*, *Gsc*, *Otx2* and *Shh*) did not affect other brain markers (*En1*, *Hoxb1* and *six3*) at E8.5 while it led to the failure of further developmental progression and later germ layer marker gene expressions (such as *Ir3*, *Nkx2.5*, *Gata6* and *Tcf15*) in E9.5 KO embryos. It seems that *Strap* deficiency leads to overall developmental arrest by affecting widespread genes throughout all three germ layers rather than specifically affecting a subset of brain specific target genes. Authors should discuss about the results in more detail.”

Response: We have discussed this concern as follow. *Strap*^{-/-} mutants failed to express several early fore- and midbrain developmental markers (i.e. *Fgf8*, *Gsc*, *Otx2* and *Shh18*) as early as E8.5, (new Fig. 2c); however, other early brain markers (i.e. *En1* and *Hoxb1* for the early hindbrain as well as *Six3* for the rostral forebrain fate), and early cardiac and endoderm lineage markers (*Gata4* and *EpCAM*) were not affected by the ablation (new Supplementary Fig. 2 c,d). In E9.5 *Strap*^{-/-} embryos, the expression of the later germ layer markers was strikingly

lower when compared to those in either Strap^{+/+} or Strap^{+/-} counterparts (new Fig. 2d). We conclude that Strap deficiency initially has a negative impact on mouse embryo fore- and midbrain development at E8.5 and subsequently cause a uniform and constant developmental delay from E9.5. As advised by the reviewer, we have discussed them in the main text of the results section (page 4).

6. Comment: “Immunocytochemistry analysis of Strap localization indicated its co-localization with several splicing factors but not with spliceosome disassemble protein DDX15 (Fig 2i and Supplementary Fig 2f). However, the immunostaining signals of STRAP and other proteins seem often over-saturated especially in the case which the authors argue the co-localization. Can the authors provide under saturated images to justify co-localizations better”?

Response: *We have replaced the saturated images with new ones in new Fig.3d.*

REVIEWERS' COMMENTS

Reviewer #1 (Remarks to the Author):

The authors have addressed my comments. The manuscript show that STRAP regulates splicing with convincing biochemical and mol bio analysis. And importantly the authors show STRAP KO leads to severe phenotypes. The paper has the makings of nature communications.

Reviewer #2 (Remarks to the Author):

None

Reviewer #3 (Remarks to the Author):

These authors have diligently answered my queries. I feel the manuscript should be accepted for publication. I just have 2 small requests that remain:

- 1) The authors have used RIP to validate eCLIP data. The text is phrased in such a way that it suggests that RIP can define the actual position of binding. I feel it would be more transparent to say that RIP was used to confirm binding to the transcript of interest (rather than the actual site of binding on the transcript)
- 2) In the supplementary data, the DNase treatment experiment doesn't add to the story and is not really referred to in the text. I suggest it should be removed and used in a follow up study

Reviewer #4 (Remarks to the Author):

In this revised manuscript, the authors satisfactorily addressed all my questions. I feel that the revised manuscript has been much improved to merit its publication in Nature communications. First of all, the quality of the experimental data (especially frog embryo experiments) has been greatly improved with the addition of new experiments and the conclusions are now better supported. Specifically, by using Tet-inducible knockdown system, the authors now clearly addressed the directness of of Strap function on AS regulation. Additional morpholino knockdown experiments in Xenopus greatly helped to strengthen their overall interpretations of the phenotypic observations and the conclusions of evolutionary conservation of Strap function.

Reviewer #1 (Remarks to the Author):

Comment: "The authors have addressed my comments. The manuscript show that STRAP regulates splicing with convincing biochemical and mol bio analysis. And importantly the authors show STRAP KO leads to severe phenotypes. The paper has the makings of nature communications."

Response: We thank the reviewer for the positive feedback.

Reviewer #2 (Remarks to the Author):

None

Reviewer #3 (Remarks to the Author):

Comment: "These authors have diligently answered my queries. I feel the manuscript should be accepted for publication. I just have 2 small requests that remain":

1) "The authors have used RIP to validate eCLIP data. The text is phrased in such a way that it suggests that RIP can define the actual position of binding. I feel it would be more transparent to say that RIP was used to confirm binding to the transcript of interest (rather than the actual site of binding on the transcript)"

Response: We agree with the reviewer that we should clearly elaborate the purpose of performing RIP assays for validating eCLIP data, instead of defining binding sites. We have revised the text accordingly.

Comment: 2) In the supplementary data, the DNase treatment experiment doesn't add to the story and is not really referred to in the text. I suggest it should be removed and used in a follow up study

Response: We thank the reviewer for pointing this out. Because TFIII complex ranks top 2 among the STRAP binding partners, reviewer #1 suggested us in the previous review to detect this complex besides the spliceosome components. We therefore performed DNase treatment experiment and clarified that STRAP interacts with TFIII C110 in a DNA-dependent manner. Here, we would like to present these data to provide an overall picture in the context of STRAP-binding proteins, as per reviewer# 1's recommendation.

Reviewer #4 (Remarks to the Author):

Comment: "In this revised manuscript, the authors satisfactorily addressed all my questions. I

feel that the revised manuscript has been much improved to merit its publication in Nature communications. First of all, the quality of the experimental data (especially frog embryo experiments) has been greatly improved with the addition of new experiments and the conclusions are now better supported. Specifically, by using Tet-inducible knockdown system, the authors now clearly addressed the directness of of Strap function on AS regulation. Additional morpholino knockdown experiments in Xenopus greatly helped to strengthen their overall interpretations of the phenotypic observations and the conclusions of evolutionary conservation of Strap function.”

Response: We thank the reviewer for the positive evaluation of our work.